# ROUTEFINDER: TOWARDS FOUNDATION MODELS FOR VEHICLE ROUTING PROBLEMS

## ABSTRACT

This paper introduces ROUTEFINDER, a comprehensive foundation model framework to tackle different Vehicle Routing Problem (VRP) variants. Our core idea is that a foundation model for VRPs should be able to represent variants by treating each as a subset of a generalized problem equipped with different attributes. We propose a unified VRP environment capable of efficiently handling any attribute combination. The ROUTEFINDER model leverages a modern transformer-based encoder and global attribute embeddings to improve task representation. Additionally, we introduce two reinforcement learning techniques to enhance multi-task performance: mixed batch training, which enables training on different variants at once, and multi-variant reward normalization to balance different reward scales. Finally, we propose efficient adapter layers that enable fine-tuning for new variants with unseen attributes. Extensive experiments on 48 VRP variants show ROUTEFINDER achieves competitive results. Our code is openly available.

## 1 INTRODUCTION

Vehicle Routing Problems (VRPs) are an important class of Combinatorial Optimization (CO) problems that have received much attention in Operations Research (OR) and Computer Science. Since the VRP is an NP-hard problem, finding an optimal solution by exhaustively exploring the solution space is often computationally expensive and impractical for large instances. Instead, heuristic methods that quickly generate good (but possibly suboptimal) solutions are commonly used. The OR community has developed many heuristics over the year, including the well-known Lin-Kernighan-Helsgaun (LKH) heuristic (Helsgaun, 2017), Fast Iterated Local Optimization (FILO) (Accorsi & Vigo, 2021; 2024) and Hybrid Genetic Search (HGS) (Vidal, 2022; Wouda et al., 2024). While these algorithms are state-of-the-art on a range of VRP variants, they often require careful consideration of the problem specifics, algorithm parameters, and computational resources to achieve the best results, and thus require considerable expert knowledge to be applied in practice.

Recently, Neural Combinatorial Optimization (NCO) approaches have been developed to solve CO problems. By leveraging deep learning, these approaches seek to learn and generalize from data, potentially providing more flexible and scalable solutions (Kool et al., 2019; Hottung & Tierney, 2019; Kwon et al., 2020; Kim et al., 2022; Berto et al., 2024; Hottung et al., 2024). In this way, optimization problems essentially become data science problems, making them more accessible.

Similar to how the developments in natural language processing have resulted in Large Language Models (LLMs), research efforts in solving CO problems through machine learning are also trending toward foundation models (Liu et al., 2024c; Ye et al., 2024a; Liu et al., 2024a; Zhou et al., 2024). However, despite the recent progress made in learning VRP variants, there is a lack of a unified approach that can effectively tackle a wide range of tasks without needing high-quality labeled datasets, which is crucial for real-world impact. Such an approach would additionally provide a platform for effectively finetuning unseen variants (Lin et al., 2024). A foundation model for VRPs would have important implications in terms of cost savings for companies and organizations as it can be easily *adapted* to new business requirements (constraints) outside of the training distribution.

In this work, we introduce ROUTEFINDER, a comprehensive foundation model framework for solving VRPs. We summarize our key contributions, including problem formulation, modeling, training, and finetuning, as follows:

- We introduce a general framework to solve different VRP variants via a unified VRP environment that can handle any number of attributes.

- We propose a modern Transformer-based architecture and introduce *Global Attribute Embeddings* to enable the model to better understand and differentiate between VRPs.

- We introduce two novel reinforcement learning techniques, *Mixed Batch Training* and *Multi-Variant Reward Normalization*, to ensure stable and effective training across multiple VRP variants.

- We present *Efficient Adapter Layers*, a lightweight yet powerful mechanism for finetuning pre-trained ROUTEFINDER models to tackle new variants with unseen attributes.

We evaluate ROUTEFINDER through extensive experiments on 48 VRP variants, assessing the impact of each novel component on performance. ROUTEFINDER significantly outperforms recent multi-task learning models by reducing optimality gaps by more than $10\%$ across all variants.

## 2 RELATED WORKS

**Neural combinatorial optimization for VRPs**    NCO has emerged as a pivotal solution approach for VRPs and other CO problems, leveraging advancements in machine learning and neural network architectures (Bengio et al., 2021; Peng et al., 2021; Mazyavkina et al., 2021; Bogyrbayeva et al., 2022). The seminal work of Vinyals et al. (2015) using pointer networks paved the way to apply these techniques to CO problems, where they now routinely find near-optimal solutions for VRPs through further developments by Bello et al. (2016) and Nazari et al. (2018). Subsequent innovations, including the transformer-based encoder with self-attention of Kool et al. (2019), POMO (Kwon et al., 2020) and Sym-NCO (Kim et al., 2022), have significantly enhanced solution generation and improvement strategies for VRPs. These advancements have been complemented by novel training algorithms, including learning with (partial) problem re-encoding at each step (Bdeir et al., 2022; Drakulic et al., 2024; Luo et al., 2024a;b) and population-based approaches (Grinsztajn et al., 2024; Hottung et al., 2024; Chalumeau et al., 2024).

Despite this progress, challenges remain in the form of requiring manual tuning for inductive bias, the need for problem-specific models, and lack of generalization, which impact deployment and generalizability (Liu et al., 2023; Thyssens et al., 2023). The field has also explored non-autoregressive solution construction methods that allow for better generalization, such as predicting promising edges (Joshi et al., 2020; Fu et al., 2021; Kool et al., 2022; Sun & Yang, 2024), improvement methods iteratively refining solutions through local adjustments or sequential rewriting (Hottung & Tierney, 2019; Ma et al., 2021; 2022; 2024), and test-time adaptation methods (Hottung et al., 2021; Choo et al., 2022) which allow for solution improvement given larger time budgets. Recent works additionally explore alternative ways of solving VRPs, such as learning heuristics for Ant Colony Optimization (Ye et al., 2024b; Kim et al., 2024) and divide-and-conquer methods (Kim et al., 2021; Li et al., 2021; Hou et al., 2022; Ye et al., 2024c; Chen et al., 2024; Zheng et al., 2024).

**Multi-task learning for VRPs**    In this work, we develop a unified VRP solver that can be generalized to any number of VRP variants. This issue of generalization has garnered much attention recently. Wang & Yu (2023) introduces a multi-armed bandit method that solves several VRP variants with limited training budgets. Lin et al. (2024) proposes training a *backbone* model (i.e., deep layers) for VRPs that can then be adapted via low-dimensional layers such as linear projections to fine-tune different problems efficiently. Drakulic et al. (2024) propose a multi-task model for CO problems trained via supervised learning, akin to LLMs. Jiang et al. (2024a) introduce UNCO, a method to transfer different problems to the embedding space via textual description through an LLM; however, UNCO still falls short in terms of performance compared to state-of-the-art NCO methods. Most related to this work are the works of Liu et al. (2024a) and Zhou et al. (2024), which use attribute composition (Ruis et al., 2021) to achieve (zero-shot) generalization on several VRP variants. Liu et al. (2024a) builds on the Reinforcement-Learning-based POMO (Kwon et al., 2020), on top of which Zhou et al. (2024) employ a mixture-of-experts model to improve generalization.

## 3 PRELIMINARIES

### 3.1 VEHICLE ROUTING PROBLEMS

We first formulate the Capacitated VRP (CVRP), the base of several more complex VRPs. The CVRP is formulated on a graph $G = (N, E)$, where $N = \{0, \ldots, m-1, m, \ldots, m+n-1\}$ represents the set of nodes, with $N_d = \{0, \ldots, m-1\}$ denoting the $m$ depots (with the classic CVRP having a single depot, i.e., $m = 1$) and the $N_c = \{m, \ldots, m+n-1\}$ denoting the $n$ customers. Each customer $i \in N_c$ has a demand $q_i$. The edges $E$ connect pairs of nodes, and each edge $(i, j) \in E$ has a travel cost $c_{ij}$ (e.g., distance or travel duration). A fleet of vehicles, each with a capacity $Q$, departs from the depot to serve each of the customers exactly once and returns, with the objective of minimizing the total travel cost. Following Vidal et al. (2014), we consider

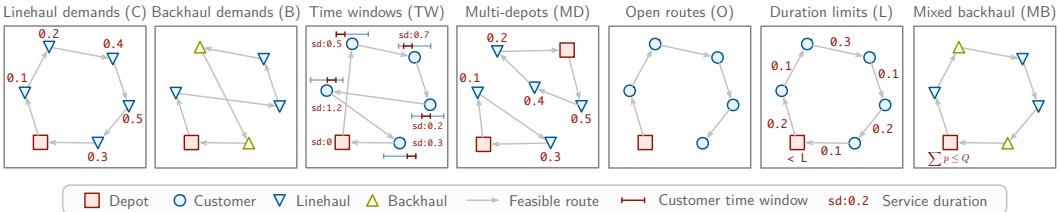

Figure 3.1: VRP attributes. Linehaul demands (C), backhaul demands (B), time windows (TW), and multi-depot (MD) are *node attributes*, whereas open routes (O), duration limits (L), and mixed backhaul (MB) mode are *global attributes*. Attribute combinations can define new VRP variants.

a collection of VRP *variants* that each consist of one or more *attributes*, resulting in a rich set of routing problems with practical relevance. Each of these variants offers a unique generalization task for ROUTEFINDER. Table A.1 in the provides a list of all 48 VRP variants we consider in this paper. We divide the attributes we consider into *node attributes*, *global attributes*, and *edge attributes*. *Node attributes* are specific to the depot and customer nodes and local to specific nodes, such as (linehaul) demands, backhaul demands, and time windows. *Global attributes* represent structural aspects of the problem as a whole, e.g., open vs. closed routes, distance limits, and the type of backhaul. In this work, the relevant *edge attribute* we consider is the cost of each edge, representing a distance. Fig. 3.1 describes the attributes modeled in this work.

NODE ATTRIBUTES

**Demand and Vehicle Capacity (C)** $[q \in [0, Q]]$: Every customer $i \in N_c$ has a linehaul demand $q_i$ that needs to be served using vehicles with a homogeneous fixed capacity $Q > 0$. The total customer demand in the vehicle must not exceed its capacity at any point of the route.

**Backhauls (B)** $[p \in [0, Q]]$: Backhauls generalize demand to also account for return shipments. Customers are either linehaul or backhaul customers. Linehaul customers require delivery of a demand $q_i$ that needs to be transported from the depot to customer $i$ (as in the CVRP), whereas backhaul customers need a pickup of an amount $p_i$ that is transported from the client back to the depot. It is possible for vehicles to serve a combination of linehaul and backhaul customers in a single route, but then any linehaul customers must precede the backhaul customers in the route. An application with returnable bottles is presented in Ropke & Pisinger (2006): full bottles need to be delivered from the depot to customers, while empty bottles are returned to the depot via backhaul.

**Time Windows (TW)** $[e, s, l \in [0, T]^3]$: Every customer $i \in N_c$ has a time window $[e_i, l_i]$ during which service must begin. Service takes $s_i$ time. The depot has a time window $[e_0, l_0] = [0, T]$, and a service duration of $s_0 = 0$. Vehicles must reach node $i$ before the end of its time window at $l_i$, but any early arrivals must wait at the node location until time $e_i$ before service may start.

**Multi-depot (MD)** $[m > 1]$: Generalizes single-depot ($m = 1$) variants as CVRP with multiple starting nodes $m > 1$ from which vehicles can their start their tour. Each vehicle must return to its start depot. This variant requires decisions about depot-customer assignments, making the problem more realistic for organizations operating from multiple facilities (Karakatič & Podgorelec, 2015).

GLOBAL ATTRIBUTES

**Open Routes (O)**  [$o \in \{0, 1\}$]: Vehicles are not required to return to the depot after serving all customers. Open routes can be found in applications with third-party drivers, who are often only compensated until they have completed their last delivery (Li et al., 2007).

**Duration Limits (L)**  [$l \in [0, L]$]: Imposes a limit on the total travel duration (or length) of each route, balancing the workload across vehicles. This limit is uniformly applied to all routes.

**Mixed Backhauls (MB)**  [$\mu \in \{0, 1\}$]: Relaxes the strict precedence constraint of linehaul customers preceding backhaul customers: with mixed backhauls, linehaul, and backhaul customers may be mixed along a route in any configuration. The vehicle's capacity must, of course, still be respected at any point along the route. Since both the current carried linehaul and backhaul demand need to be tracked for each vehicle, this variant requires careful planning.

### 3.2 LEARNING NEURAL SOLVERS FOR VRPS

**Solving VRPs using Autoregressive Sequence Generation**  Autoregressive (AR) methods address CO problems by constructing solutions sequentially. The process begins with encoding the problem instance $x$ (e.g., node and global attributes) using a trainable encoder $f_\theta$ that maps $x$ to an embedding $h = f_\theta(x)$. The solution $a$ is then decoded based on $h$ through a series of actions, where each action determines the next step in the solution based on the current partial sequence. This is achieved using a decoder $g_\theta$. The encoding and decoding process can be formalized as follows:

$$a_t \sim g_\theta(a_t | a_{t-1}, ..., a_0, h), \tag{1a}$$

$$\pi_\theta(a|x) \triangleq \prod_{t=1}^{T-1} g_\theta(a_t | a_{t-1}, ..., a_0, h), \tag{1b}$$

where $a = (a_1, ..., a_T)$ represents a feasible solution to the CO problem, $T$ denotes the steps in solution construction, and $\pi_\theta$ is the stochastic solver mapping problem instance $x$ to a solution $a$.

**Training VRP Solvers via Reinforcement Learning**  The solver $\pi_\theta$ can be trained using either supervised learning (SL) or reinforcement learning (RL). This paper focuses on RL due to its ability to train solvers independent of optimal solutions. Under the RL framework, the training objective for neural combinatorial optimization solvers is defined as:

$$\theta^* = \underset{\theta}{\operatorname{argmax}} \left[ \mathbb{E}_{x \sim P(x)} \left[ \mathbb{E}_{a \sim \pi_\theta(a|x)} [R(a, x)] \right] \right], \tag{2}$$

where $P(x)$ is the distribution of problem instances, and $R(a, x)$ represents the reward (i.e., the negative cost), associated with the solution $a$ for the given $x$. The above training problem can be tackled using various RL algorithms such as REINFORCE and its modern variants (Sutton et al., 1999; Kool et al., 2019; Kwon et al., 2020).

## 4 THE ROUTEFINDER RECIPE

ROUTEFINDER leverages attribute composition from Liu et al. (2024a); Zhou et al. (2024) to solve multiple VRP variants. Attribute composition treats different variants of the VRP as combinations of fundamental attributes from Section 3.1, using a common network to learn their representations. We go one step further than previous works and consider different combinations of attributes *within* training batches (see Section 4.3.1). Fig. 4.1 provides an overview of ROUTEFINDER's architecture.

### 4.1 UNIFIED VRP ENVIRONMENT

In previous works proposing multi-task learning across VRP variants, like MTPOMO (Liu et al., 2024a) and MVMoE (Zhou et al., 2024), the training scheme samples an instance variant (CVRP, VRPTW, etc.) out of the set of available variants during training. Every instance within that batch, therefore, is of the same problem category. This can, however, bias the optimization at each gradient step toward a specific task, potentially hindering stable and effective training for a foundation model. We thus propose to learn across problems throughout training and include problem instances of various attributes within each training batch.

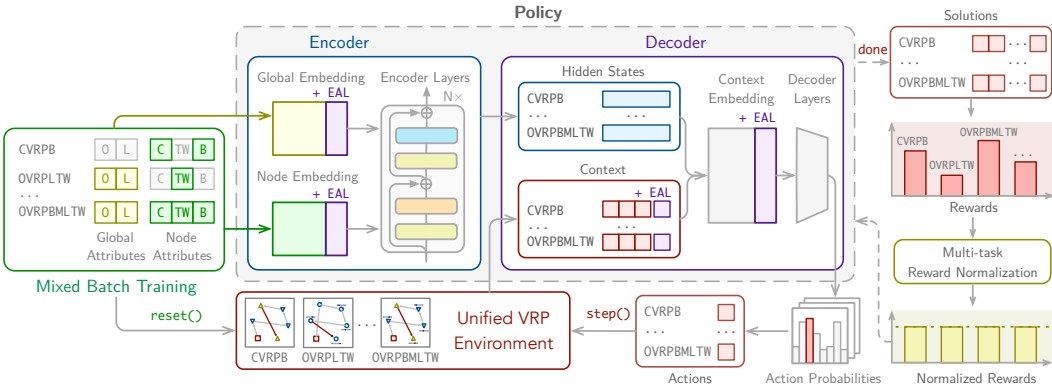

Figure 4.1: Overview of ROUTEFINDER. The unified VRP environment is used for generating data and performing rollouts (Section 4.1). Our Transformer-based encoder (Section 4.2.1) is employed to process node and global embeddings (Section 4.2.2) of problem instances. During training, we sample multiple variants in the same batch (Section 4.3.1) whose multi-task reward is then normalized (Section 4.3.2). Efficient Adapter Layers (EAL) can be employed for efficient fine-tuning to new variants (Section 4.4).

We define an environment capable of modeling all of the previously discussed VRP attributes (see Section 3.1) simultaneously, essentially building an MDOVRPMBLTW environment: a multi-depot open route vehicle routing problem with linehauls, (mixed) backhauls, distance limit, and time windows. The environment supports subsets of the MDOVRPMBLTW defining other VRP variants, i.e., some attributes can be "turned off." For example, if an instance does not have time window constraints, the time windows attribute of each customer is set to $[0, \infty]$, rendering them irrelevant during solution construction. In this way, all attributes characterizing a VRP variant can simply be turned "on" and "off", allowing us to model up to 48 different problem types with one single environment. This approach can be easily extended – for instance, by including different location sampling mechanisms and new constraints – allowing for even more future problem variants to be modeled with the same environment.

## 4.2 MODEL

### 4.2.1 TRANSFORMER-BASED ARCHITECTURE

The ROUTEFINDER transformer encoder architecture, shown in Fig. 4.2, introduces key enhancements to the standard Attention Model (AM) from Kool et al. (2019), which is the de-facto standard in recent works (Liu et al., 2024a; Zhou et al., 2024). Firstly, the ROUTEFINDER transformer encoder employs RMS (Root Mean Square) normalization (Zhang & Sennrich, 2019), improving stability and training speed by reducing the impact of outliers. Secondly, we transition from post-norm to pre-norm in transformer layers, applying normalization before the residual connections, which enhances gradient flow and promotes faster convergence (Jiang et al., 2024b). Thirdly, ROUTEFINDER uses a Feed Forward SwiGLU, (Shazeer, 2020), an extension of the Gated Linear Unit (GLU) (Dauphin et al., 2017), instead of the AM's ReLU-based feed-forward network, enhancing the model's capacity to capture complex relationships in the data. Finally, we employ FlashAttention (Dao et al., 2022; Dao, 2023) in the Multi-Head Attention layer of all models to enhance overall performance. These improvements build on recent advances in foundation models in areas such as language modeling and biology (Dubey et al., 2024; Nguyen et al., 2024), aiming to create a robust foundation model for VRPs building on modern architectures. Further details on modeling are provided in Appendix B.

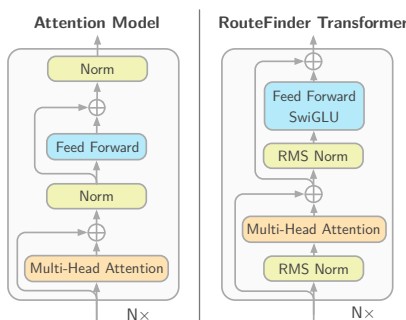

Figure 4.2: Attention model structure v.s. ROUTEFINDER transformer structure.

### 4.2.2 GLOBAL ATTRIBUTE EMBEDDINGS

Global attributes as outlined in Section 3.1 are essential for modeling VRPs; for instance, given an open (O) attribute, the solver may find optimal routes that do not necessarily loop back to the starting depot. Previous multi-task learning models for VRPs (Liu et al., 2024a; Zhou et al., 2024) project such features on the shallow decoder as dynamic features. However, such a design can be suboptimal since the deep transformer layers carry out most of the learning and, importantly, can enable effective attribute mixing, which is essential in understanding a (new) problem. We thus design Global Attribute Embeddings for effective problem representation, which incorporate problem variants and help the deep layers understand which problem is being faced. Global attributes $\phi_0, \ldots, \phi_k$ are projected via a projection layer:

$$h_g^0 = f_\theta([\phi_0, \ldots, \phi_k]), \quad f_\theta : \mathbb{R}^k \to \mathbb{R}^d \tag{3}$$

into $d$-dimensional space. Given our unified VRP representation, some attributes, such as the duration limit $l$ for unconstrained VRPs, might be $\infty$. Such attributes are padded as 0s before being processed by the deep transformer layers. We highlight the significance of Global Attribute Embeddings in Appendix D.6, where an analysis of the t-SNE latent space (Van der Maaten & Hinton, 2008) provides insights into their interpretability and importance.

### 4.3 TRAINING

### 4.3.1 VARIANT SAMPLING FOR MIXED BATCH TRAINING

Optimizing a neural solver for tackling multiple tasks requires careful consideration of its training scheme, which needs to be robust against different variant distributions. We introduce a flexible approach which we coin Mixed Batch Training (MBT) to efficiently reuse a single dataset to generate multiple problem variants, optimizing data storage and processing capabilities. We observe that the MDOVRPMBLTW problem variant is the most general problem variant we study in this paper and can be used to generate any of the other variants by selectively removing the (O), (B), (L), or (TW) attributes; for zero-shot generalization and few-shot learning, we additionally sample with the multi-depots (MD) and mixed backhaul (MB) attributes and obtain the MDOVRPMBLTW. Let $\boldsymbol{X}$ be a dataset of MDOVRPMBLTW problem instances, and let $V$ be the set of attributes, where each attribute $\nu \in V$ is associated with a sampling probability $\mathbf{p}_\nu$. For each instance $x \in \boldsymbol{X}$, we can write $x((\mathbf{1}_1)_{\nu \in V})$ to conveniently express using indicator functions $\mathbf{1}_1$ for each attribute $\nu \in V$ that the instance $x$ is equipped with $\nu$. The sampling procedure of MBT can be defined as follows:

$$\boldsymbol{X}_{\text{subsampled}} = \{x((\mathbf{1}_{\text{rand}(0,1)<\mathbf{p}_\nu})_{\nu \in V})\}_{x \in \boldsymbol{X}},$$

where $\text{rand}(0,1)$ draws an independent sample from $U[0,1]$. For example, to sample uniformly across all problem variants, we could set $\mathbf{p}_\nu = \frac{1}{2}$ for each $\nu \in V$. MBT is flexible and scalable, capable of adapting to any problem where different constraints or features might be selectively activated or deactivated. Fig. 4.3 provides an overview of MBT.

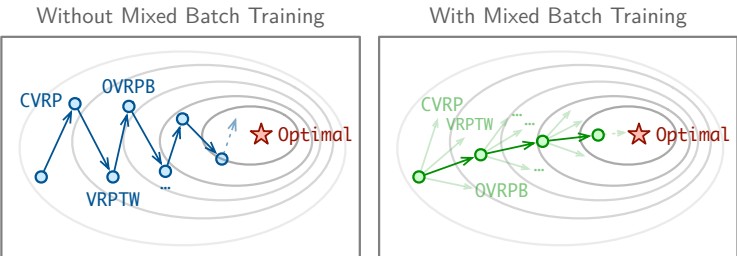

Figure 4.3: **[Left]** Training without MBT may lead to instability since at each step the optimization is biased toward a single task. **[Right]** Training ROUTEFINDER with MBT allows for more stable training.

### 4.3.2 MULTI-TASK REWARD NORMALIZATION

As explained in Section 3.2, the objective for RL-based NCO solvers is to maximize the expected reward. However, in multi-task learning settings, different problems can yield rewards on different scales. To counteract potential biases during learning, we propose to apply reward normalization per problem variant. We implement four normalization techniques to calculate the normalized re-

wards $r_{\text{norm},t}^{(k)}$ for all problem variants $k \in \{1, ..., K\}$ at training steps $t \geq 1$: 1) subtraction of the simple mean reward, 2) division through the simple mean reward, 3) subtraction of the exponentially smoothed mean, and 4) division through the exponentially smoothed mean. We calculate the average reward $\hat{r}_t^{(k)}$ *up to* training step $t$ using the average batch reward $\bar{r}_t^{(k)}$ *at training* step $t$ (see Appendix C.1). The simple mean reward at step $t$ is calculated as:

$$\hat{r}_t^{(k)} = \left((t-1) \cdot \hat{r}_{t-1}^{(k)} + \bar{r}_t^{(k)}\right) / t, \quad t \geq 1. \tag{4}$$

For the exponential moving average we set $\hat{r}_1^{(k)} = \bar{r}_1^{(k)}$ and calculate the values for $t > 1$ based on Hunter (1986) using a smoothing factor $\alpha$:

$$\hat{r}_t^{(k)} = (1-\alpha) \cdot \hat{r}_{t-1}^{(k)} + \alpha \cdot \bar{r}_t^{(k)}, \quad 0 < \alpha < 1, \quad t > 1. \tag{5}$$

The normalized rewards 1)—4) can be calculated from the original rewards $r_t^{(k)}$ according to $r_{\text{norm},t}^{(k)} = r_t^{(k)} - \hat{r}_t^{(k)}$ and $r_{\text{norm},t}^{(k)} = r_t^{(k)}/|\hat{r}_t^{(k)}|$ for subtraction and division variants, respectively. Let $\xi(\boldsymbol{a}, \boldsymbol{x}) = r_{\text{norm}}^{(k)}(\boldsymbol{a}, \boldsymbol{x})$ be a function calculating the normalized reward for instance $\boldsymbol{x}$ that additionally maps instance $\boldsymbol{x}$ to variant $k$. The multi-task reward-normalized gradient becomes:

$$\nabla_\theta J(\theta) \approx \frac{1}{N} \sum_{i=1}^{N} \left( \xi(\boldsymbol{a}^i, \boldsymbol{x}) - \frac{1}{N} \sum_{j=1}^{N} \xi(\boldsymbol{a}^j, \boldsymbol{x}) \right) \nabla_\theta \log p_\theta(\boldsymbol{a}^i|\boldsymbol{x}), \tag{6}$$

i.e., we employ the REINFORCE loss function with the POMO (Kwon et al., 2020) shared mean baseline (right side of the parenthesis) to improve convergence, where both the reward and the shared baseline are normalized by $\xi$ to calculate the policy gradient's advantage.

## 4.4 Efficient Adapter Layers: Finetuning to Unseen Attributes

Previous multi-task learning works (Liu et al., 2024a; Zhou et al., 2024) train in an environment of single-attribute VRP variants and, using compositionality (Ruis et al., 2021), achieve promising results on zero-shot generalization to VRP variants combining these individual attributes. In ROUTEFINDER, we go a step further and investigate how to efficiently generalize our pre-trained foundation model to variants with *unseen* attributes outside of the training set. Lin et al. (2024) propose pretraining a backbone model, on top of which specific Adapter Layers (AL) can be applied for more efficient finetuning to new problems – with the rationale being that the backbone (i.e., the encoder layers) may capture transferable knowledge. However, doing so excludes previous information accumulated in the projection layers from the raw attribute features to the hidden space, complicating optimization. For instance, if the first two out of $k$ dimensions encoded the Euclidean locations of nodes as $(x, y)$, re-initializing a new adapter layer from scratch will eliminate such transferable knowledge. Therefore, we propose Efficient Adapter Layers (EAL), an effective approach to learning few-shots for VRP foundation models.

Consider a linear projection layer $\mathbf{W} \in \mathbb{R}^{k \times d}$ as the original weight matrix for the projection from the raw attribute to latent space, where $k$ is the number of attributes and $d$ is the hidden dimension. In this work, for simplicity, we consider unbiased linear projections to the latent space. This can be readily extended to general affine projections using a bias term. To accommodate $l$ new attributes, EAL augments $\mathbf{W}$ with zeros. The new matrix $\mathbf{W}' \in \mathbb{R}^{(k+l) \times d}$ can be written as:

$$\mathbf{W}'^\top = \begin{bmatrix} \mathbf{W} \\ \mathbf{0} \end{bmatrix}^\top = d \left\{ \begin{bmatrix} \overbrace{\begin{matrix} w_{00} & \cdots & w_{0k} \\ \vdots & \ddots & \vdots \\ w_{d0} & \cdots & w_{dk} \end{matrix}}^{k} & \overbrace{\begin{matrix} 0 & \cdots & 0 \\ \vdots & \ddots & \vdots \\ 0 & \cdots & 0 \end{matrix}}^{l} \end{bmatrix} \right.$$

where $\mathbf{0} \in \mathbb{R}^{l \times d}$ is a matrix of zeros. The augmented matrix $\mathbf{W}'$ retains the original $k$ attributes and adds $l$ new attributes, which are initialized to zero. Doing so does not affect the model for seen attributes like AL does, as the new $l$ dimensions are "muted" until fine-tuning on new variants occurs, enabling new attributes to be included in any part of the model via EAL as shown in Fig. 4.1.

## 5 EXPERIMENTS

In this section, we empirically demonstrate the state-of-the-art performance of ROUTEFINDER in extensive experiments[1]. We address the following research questions:

**(RQ1)** Does ROUTEFINDER outperform state-of-the-art foundation models for routing problems on many different VRP variants?

**(RQ2)** How do the novel components of ROUTEFINDER contribute to its performance?

**(RQ3)** Is the proposed EAL effective in ROUTEFINDER finetuning to unseen VRP variants?

**Hardware**   All training runs are conducted on NVIDIA A100 GPUs and take between 9 to 48 hours per model. Evaluation runs are conducted on an AMD Ryzen Threadripper 3960X 24-core CPU with a single RTX 3090 GPU.

**Baselines**   *Traditional solvers*: We use PyVRP (Wouda et al., 2024), an open-source, state-of-the-art heuristic VRP solver built on top of HGS-CVRP (Vidal, 2022). PyVRP can solve all VRP variants considered in this study. We also use Google's OR-Tools (Perron & Furnon, 2023), an open-source exact and heuristic solver that relies on constraint programming and is commonly used in the ML community for its versatility to solve a large number of VRP variants. We use OR-Tools' guided local search procedure in this work. Both baseline methods solve each instance on a single CPU core with a time limit of 10 and 20 seconds for instances with 50 and 100 nodes, respectively. We parallelize traditional solvers across 16 CPU cores as in Kool et al. (2019); Zhou et al. (2024).

*Neural solvers*: We consider recent multi-task learning baselines for the VRP, including the recent MTPOMO (Liu et al., 2024a), which is based on POMO (Kwon et al., 2020), and MVMoE (Zhou et al., 2024), which introduces mixture-of-experts (Fedus et al., 2022) to improve the model performance. ROUTEFINDER variants, denoted as RF in the tables, are trained with all components proposed in the methodology section. We use Reward Normalization with division through the exponentially smoothed mean with $\alpha = 0.25$. We consider three versions of ROUTEFINDER: one version considering the (MT)POMO encoder (RF-POMO), one with the MVMoE model with four experts and hierarchical gating (RF-MoE), and one with our modern Transformer-based Encoder (RF-TE). Further details are available in Appendix B.

**Training**   We follow the setup in Kwon et al. (2020) and the recent works on MTPOMO (Liu et al., 2024a) and MVMoE (Zhou et al., 2024). Each model is trained for 300 epochs, each containing $100,000$ instances generated on the fly. We use the Adam optimizer (Kingma & Ba, 2015) with a learning rate of $3 \times 10^{-4}$ and batch size of 256. At epochs 270 and 295, the learning rate is multiplied by 0.1. Note that our setup differs from the one in Liu et al. (2024a) and Zhou et al. (2024) in that we do not artificially restrict the variants with single attributes (such as only (B) or (TW)), but train on *all* available data – similarly to how LLMs are trained on all available data, which is readily available through our unified VRP environment (more details in Appendix A).

**Evaluation**   For all ML approaches, we roll out greedy solutions using multi-starts and $8\times$ symmetric dihedral augmentations of Kwon et al. (2020), resulting in $n \times 8$ solutions per instance.

### 5.1 (RQ1) MAIN RESULTS

Table 5.1 compares ROUTEFINDER to the previously discussed baselines. We note that ROUTEFINDER variants consistently outperform other baselines across all variants by more than 10%. While changing the encoder to the MVMoE's structure (RF-MoE) may slightly improve the performance in limited settings, this comes with a higher inference cost (around 50% more) due to the more complex structure of mixture-of-experts. Conversely, the proposed Transformer Encoder (RF-TE) outperforms baselines in virtually all metrics, including low evaluation latency. Training and testing for these results are performed on the same uniform location distribution of 50 and 100 nodes; we also include results on large-scale CVRPLIB instances in Appendix D.5. Remarkably, our ROUTEFINDER does not only improve in distribution performance but can also scale better than the neural baselines in real-world settings and out-of-distribution attribute values in Appendix D.4.

---

[1]We open-source the code at: https://anonymous.4open.science/r/routefinder/

Table 5.1: Performance on 1000 test instances of trained VRPs. * represents the best-known solutions. ROUTEFINDER (RF) models improve gaps up to 20% compared to MVMoE.

| | Solver | n = 50 | | | n = 100 | | | | Solver | n = 50 | | | n = 100 | | |
|---|---|---|---|---|---|---|---|---|---|---|---|---|---|---|---|
| | | Obj. | Gap | Time | Obj. | Gap | Time | | | Obj. | Gap | Time | Obj. | Gap | Time |
| CVRP | HGS-PyVRP | 10.372 | * | 10.4m | 15.628 | * | 20.8m | VRPTW | HGS-PyVRP | 16.031 | * | 10.4m | 25.423 | * | 20.8m |
| | OR-Tools | 10.572 | 1.907% | 10.4m | 16.280 | 4.178% | 20.8m | | OR-Tools | 16.089 | 0.347% | 10.4m | 25.814 | 1.506% | 20.8m |
| | MTPOMO | 10.518 | 1.411% | 2s | 15.934 | 1.988% | 7s | | MTPOMO | 16.410 | 2.364% | 1s | 26.412 | 3.873% | 7s |
| | MVMoE | 10.501 | 1.242% | 2s | 15.888 | 1.694% | 9s | | MVMoE | 16.404 | 2.329% | 2s | 26.389 | 3.788% | 9s |
| | RF-POMO | 10.508 | 1.314% | 2s | 15.908 | 1.826% | 7s | | RF-POMO | 16.367 | 2.094% | 1s | 26.336 | 3.575% | 7s |
| | RF-MoE | **10.499** | **1.226%** | 2s | 15.876 | 1.622% | 9s | | RF-MoE | 16.389 | 2.234% | 2s | 26.322 | 3.519% | 9s |
| | RF-TE | 10.504 | 1.274% | 2s | **15.857** | **1.505%** | 7s | | RF-TE | **16.364** | **2.077%** | 1s | **26.235** | **3.178%** | 7s |
| OVRP | HGS-PyVRP | 6.507 | * | 10.4m | 9.725 | * | 20.8m | VRPL | HGS-PyVRP | 10.587 | * | 10.4m | 15.766 | * | 20.8m |
| | OR-Tools | 6.553 | 0.686% | 10.4m | 9.995 | 2.732% | 20.8m | | OR-Tools | 10.570 | 2.343% | 10.4m | 16.466 | 5.302% | 20.8m |
| | MTPOMO | 6.718 | 3.209% | 1s | 10.210 | 4.965% | 6s | | MTPOMO | 10.775 | 1.734% | 1s | 16.149 | 2.434% | 7s |
| | MVMoE | 6.702 | 2.965% | 2s | 10.177 | 4.621% | 9s | | MVMoE | 10.751 | 1.505% | 2s | 16.099 | 2.115% | 9s |
| | RF-POMO | 6.698 | 2.904% | 1s | 10.180 | 4.659% | 6s | | RF-POMO | 10.751 | 1.523% | 1s | 16.107 | 2.174% | 6s |
| | RF-MoE | 6.697 | 2.886% | 2s | 10.139 | 4.229% | 9s | | RF-MoE | **10.737** | **1.388%** | 2s | 16.070 | 1.941% | 9s |
| | RF-TE | **6.684** | **2.687%** | 1s | **10.121** | **4.055%** | 6s | | RF-TE | 10.749 | 1.502% | 1s | **16.051** | **1.827%** | 6s |
| VRPB | HGS-PyVRP | 9.687 | * | 10.4m | 14.377 | * | 20.8m | OVRPTW | HGS-PyVRP | 10.510 | * | 10.4m | 16.926 | * | 20.8m |
| | OR-Tools | 9.802 | 1.159% | 10.4m | 14.933 | 3.853% | 20.8m | | OR-Tools | 10.519 | 0.078% | 10.4m | 17.027 | 0.583% | 20.8m |
| | MTPOMO | 10.033 | 3.564% | 1s | 15.082 | 4.922% | 6s | | MTPOMO | 10.668 | 1.479% | 1s | 17.420 | 2.892% | 7s |
| | MVMoE | 10.005 | 3.270% | 2s | 15.023 | 4.508% | 8s | | MVMoE | 10.669 | 1.492% | 2s | 17.416 | 2.872% | 10s |
| | RF-POMO | 9.996 | 3.174% | 1s | 15.016 | 4.468% | 6s | | RF-POMO | 10.657 | 1.378% | 1s | 17.391 | 2.720% | 7s |
| | RF-MoE | 9.980 | 3.015% | 2s | 14.973 | 4.164% | 8s | | RF-MoE | 10.674 | 1.539% | 2s | 17.387 | 2.697% | 10s |
| | RF-TE | **9.977** | **2.989%** | 1s | **14.942** | **3.952%** | 6s | | RF-TE | **10.652** | **1.326%** | 1s | **17.327** | **2.346%** | 7s |
| VRPBL | HGS-PyVRP | 10.186 | * | 10.4m | 14.779 | * | 20.8m | VRPBLTW | HGS-PyVRP | 18.361 | * | 10.4m | 29.026 | * | 20.8m |
| | OR-Tools | 10.331 | 1.390% | 10.4m | 15.426 | 4.338% | 20.8m | | OR-Tools | 18.422 | 0.332% | 10.4m | 29.830 | 2.770% | 20.8m |
| | MTPOMO | 10.672 | 4.697% | 1s | 15.712 | 6.251% | 7s | | MTPOMO | 18.990 | 2.128% | 1s | 30.898 | 3.624% | 7s |
| | MVMoE | 10.637 | 4.354% | 2s | 15.640 | 5.758% | 9s | | MVMoE | 18.985 | 2.100% | 2s | 30.892 | 3.608% | 10s |
| | RF-POMO | 10.593 | 3.942% | 1s | 15.628 | 5.695% | 6s | | RF-POMO | **18.937** | **1.851%** | 1s | 30.796 | 3.284% | 7s |
| | RF-MoE | **10.575** | **3.765%** | 2s | 15.541 | 5.121% | 9s | | RF-MoE | 18.957 | 1.960% | 2s | 30.808 | 3.323% | 10s |
| | RF-TE | 10.578 | 3.803% | 1s | **15.528** | **5.039%** | 6s | | RF-TE | 18.941 | 1.877% | 1s | **30.688** | **2.923%** | 7s |
| VRPBTW | HGS-PyVRP | 18.292 | * | 10.4m | 29.467 | * | 20.8m | VRPLTW | HGS-PyVRP | 16.356 | * | 10.4m | 25.757 | * | 20.8m |
| | OR-Tools | 18.366 | 0.383% | 10.4m | 29.945 | 1.597% | 20.8m | | OR-Tools | 16.441 | 0.499% | 10.4m | 26.259 | 1.899% | 20.8m |
| | MTPOMO | 18.639 | 1.878% | 1s | 30.437 | 3.285% | 7s | | MTPOMO | 16.824 | 2.823% | 1s | 26.891 | 4.368% | 7s |
| | MVMoE | 18.640 | 1.883% | 2s | 30.436 | 3.281% | 9s | | MVMoE | 16.811 | 2.750% | 2s | 26.868 | 4.277% | 9s |
| | RF-POMO | 18.601 | 1.670% | 1s | 30.341 | 2.961% | 7s | | RF-POMO | **16.750** | **2.382%** | 1s | 26.783 | 3.948% | 7s |
| | RF-MoE | 18.616 | 1.757% | 2s | 30.341 | 2.954% | 9s | | RF-MoE | 16.777 | 2.550% | 2s | 26.774 | 3.912% | 9s |
| | RF-TE | **18.600** | **1.676%** | 1s | **30.241** | **2.619%** | 7s | | RF-TE | 16.762 | 2.454% | 1s | **26.689** | **3.579%** | 7s |
| OVRPB | HGS-PyVRP | 6.898 | * | 10.4m | 10.335 | * | 20.8m | OVRPBL | HGS-PyVRP | 6.899 | * | 10.4m | 10.335 | * | 20.8m |
| | OR-Tools | 6.928 | 0.412% | 10.4m | 10.577 | 2.315% | 20.8m | | OR-Tools | 6.927 | 0.386% | 10.4m | 10.582 | 2.363% | 20.8m |
| | MTPOMO | 7.108 | 3.005% | 1s | 10.878 | 5.224% | 7s | | MTPOMO | 7.112 | 3.055% | 1s | 10.884 | 5.276% | 6s |
| | MVMoE | 7.089 | 2.741% | 2s | 10.840 | 4.861% | 9s | | MVMoE | 7.098 | 2.846% | 2s | 10.847 | 4.928% | 9s |
| | RF-POMO | 7.086 | 2.688% | 1s | 10.836 | 4.821% | 7s | | RF-POMO | 7.087 | 2.693% | 1s | 10.837 | 4.830% | 6s |
| | RF-MoE | 7.080 | 2.513% | 2s | 10.805 | 4.522% | 9s | | RF-MoE | 7.083 | 2.635% | 2s | 10.806 | 4.543% | 9s |
| | RF-TE | **7.071** | **2.479%** | 1s | **10.772** | **4.208%** | 6s | | RF-TE | **7.074** | **2.508%** | 1s | **10.778** | **4.262%** | 6s |
| OVRPBLTW | HGS-PyVRP | 11.668 | * | 10.4m | 19.156 | * | 20.8m | OVRPBTW | HGS-PyVRP | 11.669 | * | 10.4m | 19.156 | * | 20.8m |
| | OR-Tools | 11.681 | 0.106% | 10.4m | 19.305 | 0.767% | 20.8m | | OR-Tools | 11.682 | 0.109% | 10.4m | 19.303 | 0.757% | 20.8m |
| | MTPOMO | 11.817 | 1.260% | 1s | 19.637 | 2.496% | 7s | | MTPOMO | 11.814 | 1.229% | 1s | 19.635 | 2.485% | 7s |
| | MVMoE | 11.822 | 1.301% | 2s | 19.641 | 2.518% | 10s | | MVMoE | 11.819 | 1.271% | 2s | 19.638 | 2.503% | 10s |
| | RF-POMO | 11.805 | 1.157% | 1s | 19.609 | 2.344% | 8s | | RF-POMO | 11.804 | 1.148% | 1s | 19.607 | 2.339% | 7s |
| | RF-MoE | 11.824 | 1.312% | 2s | 19.607 | 2.334% | 10s | | RF-MoE | 11.823 | 1.304% | 2s | 19.606 | 2.328% | 10s |
| | RF-TE | **11.805** | **1.150%** | 1s | **19.551** | **2.048%** | 7s | | RF-TE | **11.805** | **1.151%** | 1s | **19.550** | **2.042%** | 7s |
| OVRPL | HGS-PyVRP | 6.507 | * | 10.4m | 9.724 | * | 20.8m | OVRPLTW | HGS-PyVRP | 10.510 | * | 10.4m | 16.926 | * | 20.8m |
| | OR-Tools | 6.552 | 0.668% | 10.4m | 10.001 | 2.791% | 20.8m | | OR-Tools | 10.497 | 0.114% | 10.4m | 17.023 | 0.728% | 20.8m |
| | MTPOMO | 6.719 | 3.227% | 1s | 10.214 | 5.002% | 6s | | MTPOMO | 10.670 | 1.500% | 1s | 17.420 | 2.889% | 7s |
| | MVMoE | 6.707 | 3.030% | 2s | 10.184 | 4.696% | 9s | | MVMoE | 10.671 | 1.511% | 2s | 17.419 | 2.885% | 10s |
| | RF-POMO | 6.701 | 2.949% | 1s | 10.180 | 4.659% | 6s | | RF-POMO | 10.657 | 1.375% | 1s | 17.393 | 2.731% | 7s |
| | RF-MoE | 6.696 | 2.864% | 2s | 10.140 | 4.249% | 9s | | RF-MoE | 10.673 | 1.532% | 2s | 17.386 | 2.693% | 10s |
| | RF-TE | **6.686** | **2.721%** | 1s | **10.120** | **4.052%** | 6s | | RF-TE | **10.653** | **1.341%** | 1s | **17.327** | **2.347%** | 7s |

## 5.2 (RQ2) ABLATION STUDIES

We conduct ablation studies to evaluate the impact of newly introduced components. On the left of Fig. 5.1, we compare the performance of the full ROUTEFINDER (RF-TE) against its variants with ablated components, using the results for MTPOMO as a baseline. The following components are removed in the ablation studies: 1) Transformer Encoder (Section 4.2.1), 2) Global Attribute Embeddings (Section 4.2.2), 3) Mixed Batch Training (Section 4.3.1) and 4) Reward Normalization (Section 4.3.2). All components contribute to the performance of ROUTEFINDER. On the right of Fig. 5.1, we show the effect of different Reward Normalizations, i.e., 1)—4) from Section 4.3.2, with different values of $\alpha$ for the exponential moving averages. The best setting is the division through the exponentially smoothed mean with $\alpha = 0.25$. We note that future reward normalization research may further improve performance. We further provide an ablation study on the importance of the Transformer Encoder layers components in Appendix D.1 and report the effects of MBT on training stability and convergence for imbalanced variant distributions in Appendix D.2.

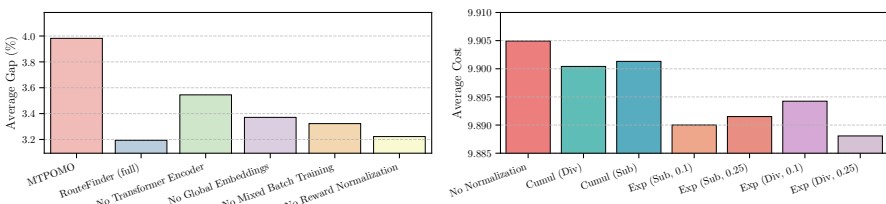

Figure 5.1: **[Left]** Ablation study on ROUTEFINDER components. **[Right]** Effect of Reward Normalization.

## 5.3 (RQ3) GENERALIZATION WITH EAL

We finally evaluate ROUTEFINDER (RF-TE) in few-shot learning settings to unseen attributes, namely the mixed (M) backhauls variants. Unlike classical backhauls, this setting allows picking up items before delivering, but the model needs to keep track of the current number of picked-up items and remaining deliverables as context and a new global attribute to learn to plan effectively. We initialize a new EAL that results in a global embedding $\mathbf{W}'_0$ adding $l = 1$ features, i.e., the mixed backhaul flag. Moreover, we encode the available load accounting for the backhaul demand picked up as a dynamic context during decoding, resulting in another EAL $\mathbf{W}'_c$, also adding one dimension. We compare against traditional baselines and 1) zero-shot performance of ROUTEFINDER, 2) training a new model from scratch, 3) AL from Lin et al. (2024), which adds new layers while keeping the pre-trained backbone, and 4) our proposed EAL. We train baselines and EAL with the same setup as the full training, but for only 10 epochs, 10K instances are sampled for each.

Table 5.2: Finetuning performance on 1000 mixed backhaul (MB) variants. ROUTEFINDER's EAL maintains the zero-shot performance and performs significantly better than other methods.

| Method | VRPMB | | OVRPMB | | VRPMBL | | VRPMBTW | | OVRPMBL | | OVRPMBTW | | VRPMBLTW | | OVRPMBLTW | |
|---|---|---|---|---|---|---|---|---|---|---|---|---|---|---|---|---|
| | Cost | Gap | Cost | Gap | Cost | Gap | Cost | Gap | Cost | Gap | Cost | Gap | Cost | Gap | Cost | Gap |
| HGS-PyVRP | 13.54 | * | 9.01 | * | 13.78 | * | 25.51 | * | 9.01 | * | 16.97 | * | 25.85 | * | 16.97 | * |
| OR-Tools | 14.93 | 10.27% | 10.59 | 17.54% | 15.42 | 11.90% | 29.97 | 17.48% | 10.59 | 17.54% | 19.31 | 13.78% | 30.44 | 17.76% | 19.31 | 13.78% |
| Zero-shot | 14.88 | 10.13% | 10.72 | 19.02% | 15.18 | 10.32% | 28.29 | 10.87% | 10.72 | 19.01% | 18.45 | 8.68% | 28.65 | 10.82% | 18.45 | 8.69% |
| Train (scratch) | 15.18 | 12.13% | 10.40 | 15.38% | 15.48 | 12.37% | 28.11 | 10.17% | 10.46 | 16.08% | 18.85 | 11.09% | 28.69 | 10.95% | 18.86 | 11.19% |
| AL (step 0) | 43.15 | 221.25% | 37.98 | 323.23% | 32.81 | 139.84% | 59.17 | 133.55% | 29.15 | 224.37% | 39.03 | 131.09% | 66.62 | 158.21% | 40.92 | 141.51% |
| AL | 14.91 | 10.10% | 10.14 | 12.53% | 15.12 | 9.73% | 27.79 | 8.92% | 10.18 | 12.95% | 18.52 | 9.13% | 28.33 | 9.56% | 18.51 | 9.05% |
| EAL (step 0) | 14.88 | 10.13% | 10.72 | 19.02% | 15.18 | 10.32% | 28.29 | 10.87% | 10.72 | 19.01% | 18.45 | 8.68% | 28.65 | 10.82% | 18.45 | 8.69% |
| EAL | **14.59** | **7.89**% | **9.66** | **7.19**% | **14.78** | **7.39**% | **26.69** | **4.61**% | **9.65** | **7.13**% | **17.60** | **3.70**% | **27.13** | **4.90**% | **17.59** | **3.65**% |

Table 5.2 shows that EAL consistently outperforms baselines in few-shot learning, with strong performance further supported by multi-depot experiments Appendix D.3. We additionally compare AL and EAL at "step 0", i.e., after replacing the new adapter layers. Notably, while AL with the untrained new layers can greatly degrade the performance unless optimization is performed, EAL maintains the zero-shot performance even without training, providing a much better starting point.

## 6 CONCLUSION

In this work, we presented ROUTEFINDER, a comprehensive framework to develop foundation models for VRPs. We introduced a unified VRP environment to represent any combination of attributes. We proposed a new Transformer Encoder and Global Attribute Embeddings to enhance learning representations of diverse VRPs. We introduced Mixed Batch Training and Multi-variant Reward Normalization to allow for effective training with RL in a multi-task setting with different tasks and reward scales. Finally, we introduced Efficient Adapter Layers, a lightweight and powerful technique to finetune ROUTEFINDER to unseen attributes. Our extensive evaluations on 24 VRP variants showed ROUTEFINDER outperforms SOTA neural baselines for VRPs.

ROUTEFINDER represents an early attempt to learn a foundation model across problem variants. While demonstrating strong generalization, it does so at a slight expense in solution quality compared to techniques trained on specific problem variants, at least for in-distribution results, as also noted by prior works (Liu et al., 2024a; Zhou et al., 2024). For future work, we intend to extend ROUTEFINDER to support further variants of the vast VRP literature. We also intend to improve the model performance to eventually outperform state-of-the-art traditional OR solvers – exciting directions include decomposition methods (Ye et al., 2024c; Zheng et al., 2024) and end-to-end construction and improvement (Kong et al., 2024).

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

## A  UNIFIED VRP ENVIRONMENT DETAILS

We consider the seven attributes from Section 3.1 for instance generation through our environment definition explained in Section 4.1. Leveraging our environment's modular structure, we build the 16 VRP variants as used in MVMoE (Zhou et al., 2024), but by differentiating between *traditional* (B) and *mixed* (MB) backhauls, as defined in Avci & Topaloglu (2015), we extend that number to 24. By considering multi-depot problems, we further increase that number to 48 variants that can be solved with ROUTEFINDER (see Table A.1).

We describe additional details of the Unified VRP environment, including data generation in Appendix A.1 and environment logic in Appendix A.2. For a better understanding, we invite the reader to look at the source code, which we tried our best to comment on for clarity, at https://anonymous.4open.science/r/routefinder/.

### A.1  DATA GENERATION

We now explain the individual steps in the data generation process we use for our modular VRP environment, including the node attributes and global attributes. While throughout the main part of this paper, we have focused on routing problems with a single depot, our unified environment can actually handle problems with multiple depots, where we define $m$ as the number of depots. For comparability to the neural baselines, the main experiments were run on single-depot problems, but we report results for multi-depot problems (Appendix D.3).

**Locations**   We generate $m + n$ locations randomly with $x_i$ and $y_i \sim U(0, 1), \forall i \in \{0, ..., m + n - 1\}$, where $[x_i, y_i], i \in \{0, ..., m - 1\}$ denote the $m$ depots and $[x_i, y_i], i \in \{m, ..., m + n - 1\}$, the $n$ customer nodes. Note that this setting can be expanded to consider more realistic distributions as in (Bi et al., 2022; Zhou et al., 2023; Gao et al., 2024), and our implementation is already set up in such a way to allow for different distributions in the future via the get_sampler method.

**Multiple depots (MD)**   Depot nodes, in principle, have the same node attributes as customer nodes. The location, however, is the only attribute that is generated in the same way, which is explained below. For all other attributes, the values are fixed and identical for all depots. Linehaul and backhaul demands, as well as service durations, are set to zero, while the time windows of all depots in an instance are set to $[e_i, l_i] = [0, t_{max}], i \in \{0, ..., m - 1\}$, where $t_{max}$ denotes the system end time and $M$ the number of depots. For problems without time windows, $t_{max}$ is set to $\infty$. In the unseen variants experiments of Appendix D.3, we employ $m = 3$ depots for the MD finetuning variants.

**Vehicle capacity (C)**   The vehicle capacity $C$ is a fixed value applied to all vehicles and calculated according to:

$$C = \begin{cases} 30 + \left\lfloor \frac{1000}{5} + \frac{n-1000}{33.3} \right\rfloor & \text{if } 1000 < n \\ 30 + \left\lfloor \frac{n}{5} \right\rfloor & \text{if } 20 < n \leq 1000 \\ 30 & \text{otherwise} \end{cases}$$

which is commonly used in NCO for VRP approaches (Kool et al., 2019; Kwon et al., 2020).

**Linehaul and backhaul demands (C) / (B) / (MB)**   We generate demands according to the following scheme:

1. Generate linehaul demands $q_i$ for all customers $i \in N_c$ by sampling uniformly from the set of integers $\{1, 2, ..., 9\}$.

2. Generate backhaul demands $p_i$ for all customers $i \in N_c$ by sampling uniformly from the set of integers $\{1, 2, ..., 9\}$.

Table A.1: The 48 VRP variants we consider. All variants include the base Capacity (C). The $k = 5$ features O, B, L, TW, and MD can be combined into any subset, including the empty set and itself (i.e., a *power set*) with $2^k = 32$ possible combinations. The Mixed (M) global feature creates new Mixed Backhaul (MB) variants in generalization studies, adding 16 more variants.

| VRP Variant | Capacity (C) | Open Route (O) | Backhaul (B) | Mixed (M) | Duration Limit (L) | Time Windows (TW) | Multi-depot (MD) |
|---|---|---|---|---|---|---|---|
| CVRP | ✓ | | | | | | |
| OVRP | ✓ | ✓ | | | | | |
| VRPB | ✓ | | ✓ | | | | |
| VRPL | ✓ | | | | ✓ | | |
| VRPTW | ✓ | | | | | ✓ | |
| OVRPTW | ✓ | ✓ | | | | ✓ | |
| OVRPB | ✓ | ✓ | ✓ | | | | |
| OVRPL | ✓ | ✓ | | | ✓ | | |
| VRPBL | ✓ | | ✓ | | ✓ | | |
| VRPBTW | ✓ | | ✓ | | | ✓ | |
| VRPLTW | ✓ | | | | ✓ | ✓ | |
| OVRPBL | ✓ | ✓ | ✓ | | ✓ | | |
| OVRPBTW | ✓ | ✓ | ✓ | | | ✓ | |
| OVRPLTW | ✓ | ✓ | | | ✓ | ✓ | |
| VRPBLTW | ✓ | | ✓ | | ✓ | ✓ | |
| OVRPBLTW | ✓ | ✓ | ✓ | | ✓ | ✓ | |
| VRPMB | ✓ | | ✓ | ✓ | | | |
| OVRPMB | ✓ | ✓ | ✓ | ✓ | | | |
| VRPMBL | ✓ | | ✓ | ✓ | ✓ | | |
| VRPMBTW | ✓ | | ✓ | ✓ | | ✓ | |
| OVRPMBL | ✓ | ✓ | ✓ | ✓ | ✓ | | |
| OVRPMBTW | ✓ | ✓ | ✓ | ✓ | | ✓ | |
| VRPMBLTW | ✓ | | ✓ | ✓ | ✓ | ✓ | |
| OVRPMBLTW | ✓ | ✓ | ✓ | ✓ | ✓ | ✓ | |
| MDCVRP | ✓ | | | | | | ✓ |
| MDOVRP | ✓ | ✓ | | | | | ✓ |
| MDVRPB | ✓ | | ✓ | | | | ✓ |
| MDVRPL | ✓ | | | | ✓ | | ✓ |
| MDVRPTW | ✓ | | | | | ✓ | ✓ |
| MDOVRPTW | ✓ | ✓ | | | | ✓ | ✓ |
| MDOVRPB | ✓ | ✓ | ✓ | | | | ✓ |
| MDOVRPL | ✓ | ✓ | | | ✓ | | ✓ |
| MDVRPBL | ✓ | | ✓ | | ✓ | | ✓ |
| MDVRPBTW | ✓ | | ✓ | | | ✓ | ✓ |
| MDVRPLTW | ✓ | | | | ✓ | ✓ | ✓ |
| MDOVRPBL | ✓ | ✓ | ✓ | | ✓ | | ✓ |
| MDOVRPBTW | ✓ | ✓ | ✓ | | | ✓ | ✓ |
| MDOVRPLTW | ✓ | ✓ | | | ✓ | ✓ | ✓ |
| MDVRPBLTW | ✓ | | ✓ | | ✓ | ✓ | ✓ |
| MDOVRPBLTW | ✓ | ✓ | ✓ | | ✓ | ✓ | ✓ |
| MDVRPMB | ✓ | | ✓ | ✓ | | | ✓ |
| MDOVRPMB | ✓ | ✓ | ✓ | ✓ | | | ✓ |
| MDVRPMBL | ✓ | | ✓ | ✓ | ✓ | | ✓ |
| MDVRPMBTW | ✓ | | ✓ | ✓ | | ✓ | ✓ |
| MDOVRPMBL | ✓ | ✓ | ✓ | ✓ | ✓ | | ✓ |
| MDOVRPMBTW | ✓ | ✓ | ✓ | ✓ | | ✓ | ✓ |
| MDVRPMBLTW | ✓ | | ✓ | ✓ | ✓ | ✓ | ✓ |
| MDOVRPMBLTW | ✓ | ✓ | ✓ | ✓ | ✓ | ✓ | ✓ |

3. For each customer $i \in N_c$, generate a temporary decision variable $z_i \in \{0, 1\}$ with probabilities $\mathbb{P}(z_i = 0) = 0.8$ and $\mathbb{P}(z_i = 1) = 0.2$.

   - If $z_i = 0$, keep the linehaul demand $q_i$ and set the backhaul demand $p_i = 0$.
   - If $z_i = 1$, set the linehaul demand $q_i = 0$ and keep the backhaul demand $p_i$.

This demand generation scheme ensures that each customer has either a linehaul demand or a backhaul demand, but not both. With a probability of 0.8, a customer will have only a linehaul demand, and their backhaul demand will be set to 0. Conversely, with a probability of 0.2, a customer will have only a backhaul demand, and their linehaul demand will be set to 0. It is important to note that not all customers are typically backhaul customers, even in a backhaul setting. Therefore, this scheme allows for the consideration of both linehaul and backhaul demands in backhaul problem settings while ensuring that each customer has only one type of demand.

We note that this can be easily extended to the case of VRP with simultaneous pickup and delivery (VRPSPD), in which a customer can have both linehaul and backhaul demand (Ai & Kachitvichyanukul, 2009; Koç et al., 2020). In such a case, we could duplicate the customer node into two nodes with the same attributes, such as locations, but different values for linehaul (pickup) and backhaul (delivery) in the current VRP environment or allow for both linehaul and backhaul to be present at the same time in a single node with small modifications in the action masking.

**Backhaul class (B) / (MB)**  For testing the few-shot setting described in Section 5.3, we generate instances with *mixed* backhauls. The instances themselves are actually identical to instances with the *traditional* backhaul, and we use a global attribute in the instance to differentiate between them. For this purpose, we allow either setting a fixed value $\in \{1, 2\}$ or sampling from $\{1, 2\}$ for every customer with equal probabilities $p(1) = p(2) = 0.5$, allowing for different backhaul settings within one batch, if needed (see the batching procedure described in Section 4.3.1). Note that we sample from $\{1, 2\}$ instead of boolean sampling because we plan to extend the number of backhaul settings in the future.

**Open routes (O)**  For open routes, we generate a boolean vector with all `True` values. During sampling (see Section 4.3.1), the actual ratio of open route instances is defined, not at the initial instance generation (i.e., we temporarily change the `True` value to `False` for every batch element with a certain probability).

**Time Windows (TW)**  We generate the time windows $[e_i, l_i]$ and service times $s_i$ in several steps for all customers $i \in N_c$:

1. Generate service times $s_i \in [0.15, 0.18]$.

2. Generate time window lengths $t_i \in [0.18, 0.2]$.

3. Calculate the maximum distance from any of the depots $j \in \{0, ..., m-1\}$ to customer $i$: $d_{max} = \max_j(d_{ij})$.

4. Calculate upper bounds for time window start times $h_i = \frac{t_{max} - s_i - t_i}{d_{max}} - 1$.

5. Calculate time window start times as $e_i = (1 + (h_i - 1) \cdot u_i) \cdot d_{max}$ with $u_i \sim U(0, 1)$.

6. Calculate time window end times as $l_i = e_i + t_i$.

When calculating the action mask, we have the constraint that the expected arrival time should be earlier than the end time of nodes; if the problem is a closed problem, we should also consider the time back to the depot, i.e., $\max(t_{\text{curr}} + d_{ij}, e_j) + s_j + d_{max} < l_0$. We note that for simplicity, we set the vehicle speed to 1.0 in equations and normalize time windows accordingly so that travel time from two nodes is the same numerically as the distance between them. This can be easily modified in the code.

We mention as an alternative TW generation procedure the one from the Solomon benchmark (Solomon, 1987; Li et al., 2021), which may perform better in that benchmark, as done in Zhou et al. (2024).

**Distance limit (L)**  The distance limit is sampled from a uniform distribution to ensure meaningful and feasible constraints. Specifically, we sample $L$ from $U(2 \cdot \max(d_{0i}), l_{\max})$, where $d_{0i}$ is the distance from the depot to customer $i$, and $l_{\max} = 3.0$ is a predefined upper bound. This approach ensures that $L$ is always greater than the round trip to the farthest customer $(2 \cdot \max(d_{0i}))$, making all customers reachable, while also allowing for variation in the constraint tightness. For the multi-depot case we replace $\max(d_{0i})$ with $\min_j(\max_i(d_{ij})), i \in \{m, ..., m + n\}, j \in \{0, ..., m\}$, i.e., we first get the maximum distance from any customer node to each of the depots and then take the minimum out of those distances. By taking the maximum in the first step we ensure that all customers are reachable, and by taking the minimum across depots, we make the problem more challenging, because even though all nodes can in principle be serviced, some may only be serviced by one (or a subset) of the available depots. This sampling method produces more variation than previous works Liu et al. (2024a); Zhou et al. (2024) (where there was virtually no difference in solutions of (L) and non-(L) variants), as it guarantees feasible instances while still providing a range of challenging scenarios.

**Attribute Normalization and Scaling**   All demands, both linehauls and backhauls, are scaled to lie in $[0, 1]$ through division by the vehicle capacity. $q_i' = q_i/C, p_i' = p_i/C$. All other features are already sampled from a normalized range. Note that during loading instances from e.g. CVRPLib, we normalize features before passing them to the policy - for instance, locations are normalized between 0 and 1.

## A.2   ENVIRONMENT LOGIC

To determine available actions for the Unified VRP environment formulation, the constraints for the individual problems have to be combined in the action mask (`action_mask` in the code following RL4CO, where `True` means that the action is feasible (Berto et al., 2024)). We build a logical test structure, essentially separating the checks in the action mask according to the individual VRP problem types and then bringing them all together again. The individual `action_mask` checks are the following:

a) *Can reach in time*: depending on the current time and the travel distance to every node not yet visited, can we reach that node before its service time window ends? $t_{\text{curr}} + d_{ij} < l_j$, where $t_{\text{curr}}$ is the current time.

b) *Does not exceed distance limit*: depending on the current length of the route, if we travel to any available node, will we exceed the total distance limit for the route? $l_{\text{curr}} + d_{ij} < L$, where $l_{\text{curr}}$ is the current length.

c) *Can reach depot*: there are two types of constraints from time windows (TW) and distance limit (L):

- If we need to ensure we can reach the depot in time, i.e., the current time plus traveling time to the depot must be smaller than the system end time: $\max(t_{\text{curr}} + d_{ij}, e_j) + s_j + d_{j0} < t_{max}$.
- If we need to ensure we can reach the depot without exceeding the distance limit, i.e., the current distance plus the traveling distance to the depot must be smaller than the distance limit: $l_{\text{curr}} + d_{ij} + d_{j0} < L$.

For the multi-depot case we replace $d_{j0}$ in both these constraints with $d_{jk}$, where $k \in \{0, ..., m - 1\}$ indexes the depot the current route *started* from. For open routes, this will always be set to `True`, i.e., this constraint does not apply.

d) *Demand constraints for backhaul problems*:

- Checks for *all* backhauls problems:
  - Does the linehaul demand exceed vehicle capacity if we add a node's demand to the current vehicle? $c_{\text{curr}} + q_j < C$, where $c_{\text{curr}}$ is the used capacity.
  - Does the backhaul demand exceed vehicle capacity if we add a node's demand to the current vehicle? $c_{\text{curr}} + p_j < C$, where $c_{\text{curr}}$ is the used capacity.
- Checks for traditional backhaul settings:
  - Carrying backhaul: if we are already picking up backhaul demands, we cannot service any linehaul demands on this route anymore.
  - If we are not carrying backhaul demands yet, are there any unserved linehaul demands left?
  - If there are no linehaul demands left or we are already carrying backhauls, are there still unserved backhaul demands?
- Checks for *mixed* backhaul settings:
  - Cannot service linehaul demands: depending on the backhaul demands currently loaded in the vehicle, do we have space left for further linehaul demands?

  We additionally remark that our definition of backhauls follows the generally accepted definition in the OR community, originally due to Goetschalckx & Jacobs-Blecha (1989). This definition differs from the routing problems with backhaul considered in several recent papers in the machine learning (e.g., Liu et al. (2024a); Zhou et al. (2024)), who define backhaul customers as having a negative demand of the same commodity used for linehaul, and do not consider the precedence constraint that all linehaul must be completed before backhaul may start on the route. The problem setting with a single commodity is not commonly

studied in the OR literature since it implies pickups may be used for deliveries at later customers, while the relaxation of the precedence constraint is more properly referred to as a *mixed* backhaul problem (Koç & Laporte, 2018).

e) *Already visited*: every customer node needs to be visited exactly once.

We bring together checks a) to e) and introduce an additional check for the depot: if we are currently in the depot and there are still unserved customers, we cannot select the depot as the next action to ensure the model cannot get stuck during decoding. For the multi-depot case we further extend this check. If we are currently in a depot and there are unserved customers, we cannot visit *any* depot. If no further customers can be serviced, all depots are available actions again. However, if we are currently in a depot and no customers can be served from this depot, we mask it out so as to service the remaining customers from the remaining depots that can actually service them.

Combining these checks in this way allows us to meticulously check for individual VRP settings while at the same time maintaining the necessary flexibility the unified environment formulation requires.

## B  ROUTEFINDER MODEL DETAILS

ROUTEFINDER follows the encoder-decoder architecture from the Attention Model (Kool et al., 2019), a transformer-like architecture based on the attention mechanism (Vaswani et al., 2017). We additionally improve the encoder architecture in RF-TE as explained in Section 4.2. We focus the explanation on modeling *all* attributes possible with the MDOVRPMBLTW, noting that in the main training runs, we do so without considering attributes from multi-depots and mixed backhaul, whose additional parameters are added upon EAL finetuning.

### B.1  MULTI-HEAD ATTENTION

At the core of ROUTEFINDER lies the Multi-Head Attention (MHA) mechanism, proposed by Vaswani et al. (2017). MHA concurrently attends to information from various representation subspaces, facilitating the capture of diverse relationships between input elements. Notably, MHA is capable of handling a variable number of elements.

The MHA operation starts by linearly projecting the input sequences of queries $Q$, keys $K$, and values $V$ to $H$ distinct subspaces using learned projection matrices $W_i^Q$, $W_i^K$, and $W_i^V$, respectively, where $H$ denotes the number of attention heads: $Q_i = QW_i^Q$, $K_i = KW_i^K$, $V_i = VW_i^V$ for $i = 1, \ldots, H$. Subsequently, the attention weights for each head are computed by performing a scaled dot product between the projected queries and keys, followed by a softmax operation:

$$A_i = \text{Softmax}\left(\frac{Q_i K_i^T}{\sqrt{d_k}} + M\right) \tag{7}$$

where $d_k$ represents the dimension of the keys, acting as a scaling factor to prevent the dot products from growing too large, $\text{Softmax}(x_i) = \frac{\exp(x_i)}{\sum_{j=1}^{N} \exp(x_j)}$ and $M$ is an optional attention mask that can be used to prevent attending to certain positions (e.g., infeasible actions), which can be done by setting elements to $-\infty$. The output of each attention head is then calculated as a weighted sum of the projected values, using the attention weights: $Z_i = A_i V_i$.

Lastly, the outputs from all attention heads are concatenated and linearly projected using a learned matrix $W^O$ to yield the final output of the MHA operation:

$$\text{MHA}(Q, K, V) = \text{Concat}(Z_1, \ldots, Z_H)W^O \tag{8}$$

While the MHA grows quadratically, i.e., with sequence length (i.e., number of nodes) $N$, it grows as $O(N^2)$, several efficient implementations have been proposed over the years, and we use FlashAttention (Dao et al., 2022; Dao, 2023) to speed up the model.

### B.2  ENCODER

The Encoder transforms an input instance $\boldsymbol{x}$ into a hidden embedding $\boldsymbol{h}$. The Encoder architecture consists of the following main components: 1) Global Embedding, 2) Node Embedding, and 3)

a series of Encoder Layers. We consider a VRP instance of $n$ locations as having $n + 1$ nodes, where node $0$ is the depot and nodes $\{1, \ldots, n\}$ are $n$ customers. For problems with multiple depots, we define $m$ as the number of depots, i.e., nodes $\{0, \ldots, m - 1\}$ are the depot nodes, and $m, \ldots, m + n - 1$ are the $n$ customer nodes.

**Global Embedding** Since Global Attributes contain a single value for all the $m + n$ problem nodes, we embed them in depot nodes, in a similar fashion to how traditional solvers as PyVRP encode information about the global problem structure on depot nodes.. Global Embeddings include global attributes Open Routes $o \in \{0, 1\}$, Duration Limits $l \in [0, L]$, and Mixed Backhauls flag $\mu \in \{0, 1\}$, as well as the locations of the depot node(s) $[x_i, y_i] \in \mathbb{R}^2, i \in \{0, \ldots, m - 1\}$ and the system end time $l_{\max}$ (i.e., the depot(s) time window). In practice, for the multi-depot case with $m > 1$, the global attributes are projected on the depot nodes. In ROUTEFINDER, the global embedding $f$ is a linear projection layer $\mathbf{W}_g \in \mathbb{R}^{k \times d}$ where $k = 6$ features and $d = 128$ is the hidden dimension. The initial projected global hidden embedding per depot $g_i$ can be written as $\boldsymbol{h}_{g_i}^{(0)} = \mathbf{W}_g [x_i, y_i, l_{\max}, o, l, \mu]^\top$.

**Node Embedding** The node embeddings, on the other hand, capture customer-specific attributes and are projected onto the remaining $n$ nodes. These attributes include for nodes $i \in \{m, \ldots m + n - 1\}$: Linehaul demands $q_i \in [0, Q]$, Time Windows parameters $e_i, s_i, l_i \in [0, T]^3$ where $e$ and $l$ denote the time window's start and end and $s$ is the service time, the Backhaul demands $p_i \in [0, Q]$, and finally the node locations $[x_i, y_i] \in \mathbb{R}^2$. In ROUTEFINDER this a linear projection layer $\mathbf{W}_n \in \mathbb{R}^{k \times d}$ where $k = 7$ features and $d = 128$ is the hidden dimension. The initial projected node hidden embedding can be written for each node $n_i$ as $\boldsymbol{h}_{n_i}^{(0)} = \mathbf{W}_n [x_i, y_i, q_i, e_i, s_i, l_i, p_i]^\top$.

**Raw Features to Hidden States** The projected global embedding and node embeddings are concatenated to obtain the initial hidden representation $\boldsymbol{h}^{(0)} \in \mathbb{R}^{(m+n) \times d}$, where $m + n$ is the total number of nodes ($m$ depots + $n$ customers) and $d$ is the hidden dimension:

(9)

$$bmh^{(0)} = \text{Concat}(\boldsymbol{h}_{g_1}^{(0)}, \ldots, \boldsymbol{h}_{g_m}^{(0)}, \boldsymbol{h}_{n_1}^{(0)}, \ldots, \boldsymbol{h}_{n_n}^{(0)}) \tag{10}$$

The initial hidden representation $\boldsymbol{h}^{(0)}$ is then passed through a series of Encoder Layers to refine and enrich the representation. Each Encoder Layer consists of a Multi-Head Attention (MHA) layer and a Multi-Layer Perceptron (MLP) layer, as described in Eq. (12) and Eq. (13), respectively.

The Encoder can be represented as:

$$\boldsymbol{h} = \text{EncoderBlocks}(\boldsymbol{h}^{(0)}) \tag{11}$$

Each EncoderBlock consists of two sub-layers: a Multi-Head Attention (MHA) layer and a Multi-Layer Perceptron (MLP) layer (or SwiGLU as we propose). The MHA layer allows the model to capture dependencies between different positions in the input sequence, while the MLP layer applies non-linear transformations to the features at each position. The input to each EncoderBlock is first passed through the MHA layer, which computes the self-attention using the input as queries, keys, and values:

$$\hat{\boldsymbol{h}} = \text{Norm}\left(\boldsymbol{h}^{(\ell-1)} + \text{MHA}(\boldsymbol{h}^{(\ell-1)}, \boldsymbol{h}^{(\ell-1)}, \boldsymbol{h}^{(\ell-1)})\right) \tag{12}$$

where $\boldsymbol{h}^{(\ell-1)}$ represents the input to the $\ell$-th EncoderBlock, and Norm denotes a normalization operation, in ROUTEFINDER we employ Instance Normalization (IN). The output of the MHA layer, $\hat{\boldsymbol{h}}$, is then passed through the MLP layer, which applies a series of linear transformations with non-linear activations:

$$\boldsymbol{h}^{(\ell)} = \text{Norm}\left(\hat{\boldsymbol{h}} + \text{MLP}(\hat{\boldsymbol{h}})\right) \tag{13}$$

The pointwise MLP layer consists of two linear layers with a non-linear activation function as ReLU, between them.

**Transformer-based Encoder** We further explicit our proposed Transformer-based encoder. Each EncoderBlock consists of two sub-layers: a Multi-Head Attention (MHA) layer and a Feed Forward SwiGLU layer (Shazeer, 2020). The MHA layer captures dependencies between different positions in the input sequence, while the SwiGLU layer applies non-linear transformations to the features. We employ RMS normalization (Zhang & Sennrich, 2019) and pre-norm architecture for improved

stability and faster convergence:

$$\hat{\boldsymbol{h}} = \boldsymbol{h}^{(\ell-1)} + \text{MHA}(\text{RMSNorm}(\boldsymbol{h}^{(\ell-1)}), \text{RMSNorm}(\boldsymbol{h}^{(\ell-1)}), \text{RMSNorm}(\boldsymbol{h}^{(\ell-1)})) \quad (14)$$

$$\boldsymbol{h}^{(\ell)} = \hat{\boldsymbol{h}} + \text{SwiGLU}(\text{RMSNorm}(\hat{\boldsymbol{h}})) \quad (15)$$

where $\boldsymbol{h}^{(\ell-1)}$ represents the input to the $\ell$-th EncoderBlock. The SwiGLU activation function is defined as:

$$\text{SwiGLU}(\boldsymbol{x}) = \boldsymbol{x} \odot \sigma(\boldsymbol{W}_1 \boldsymbol{x} + \boldsymbol{b}_1) \otimes \text{SiLU}(\boldsymbol{W}_2 \boldsymbol{x} + \boldsymbol{b}_2) \quad (16)$$

where $\odot$ denotes element-wise multiplication, $\otimes$ is matrix multiplication, $\sigma$ is the sigmoid function, SiLU is the Sigmoid Linear Unit (Swish) activation function, and $\boldsymbol{W}_1, \boldsymbol{W}_2, \boldsymbol{b}_1, \boldsymbol{b}_2$ are learnable parameters. We use FlashAttention (Dao et al., 2022; Dao, 2023) in the MHA layer for enhanced performance.

### B.3 DECODER

The Decoder autoregressively constructs the solution based on the Encoder output $\boldsymbol{h}$ and the state $s_t$ at the current step $t$.

**Context Embedding** The context embedding is used to modify the query embedding of the problem node of the current partial solution. It consists of a linear layer that projects the concatenated current node embedding and state embedding to the embedding space. The state embedding is computed by projecting the following: the current node embedding $\boldsymbol{h}_t$ and a set of dynamic features from state $s_t$, i.e. the available load $c_t$, current time $t_t$, current distance traveled $d_t$, the available backhaul load $b_t$ – i.e. the difference between the vehicle capacity $Q$ and the *used backhaul capacity*, which is necessary because if we pick up items, the deliverable quantity must exceed the remaining capacity after pick up for mixed backhauls (MB) – as well as the location of the origin depot $o$ we have to return to at step $t$: $[x_t^o, y_t^o]$ for the multi-depot variants (MD). In ROUTEFINDER the context embedding $\mathbf{W}_c \in \mathbb{R}^{d \times (d+k)}$ is a linear projection matrix, $d = 128$ is the hidden dimension, and $k = 6$ is the number of state features. The context embedding at step $t$ is thus computed as $\mathbf{h}_c^{(t)} = \mathbf{W}_c \text{Concat}([\boldsymbol{h}_t; [c_t, t_t, d_t, b_t, x_t^o, y_t^o]])^\top$.

**Attention and Pointer Mechanism** The query $q_t$ is obtained directly from the context embedding $q_t = \mathbf{h}_c^{(t)}$ and then passed into a masked MHA layer and final single-head attention to obtain logits $\boldsymbol{z}$:

$$h_t^c = \text{MHA}(q_t, K_t^g, V_t^g, M_t), \quad (17)$$

$$\boldsymbol{z} = \frac{V_t^p h_t^c}{\sqrt{d_k}} \quad (18)$$

where $M_t$ is the set of feasible actions (i.e., the `action_mask`), and projections $K_t^g, V_t^g, V_t^p = W_k^g \boldsymbol{h}, W_v^g \boldsymbol{h}, W_v^p \boldsymbol{h}$ are precomputed once as cache. We note that Eq. (18) is usually referred to as the pointer mechanism (Vinyals et al., 2015).

**Logits processing** Finally, logits $\boldsymbol{z}$ are transformed into a probability distribution:

$$p = \text{Softmax}\left(C \cdot \tanh(\boldsymbol{z})\right) \quad (19)$$

where logits for infeasible actions can be masked, and $C$ is the *tanh clipping* that serves in improving the exploration, which we set to 10 according to Bello et al. (2016).

**Action selection** During training, we use the POMO `multistart` sampling. For the multi-depot case we force the first action to start from all depots in the instance. For the single-depot case we force the first action to start with every customer node to maximize diversity. Note that if `num_starts` is not divisible by the number of depots $m$, the resulting tensor will not have an equal number of indices for each depot, i.e., the number of starts will not be distributed evenly across the depots, as we use the modulo operator for the assignment.

During testing, we also employ `multistart` but with greedy selection (i.e., selecting the maximum probability). Prior to the selection, a dihedral augmentation is also performed prior to encoding instance $\boldsymbol{x}$ in the encoder, which enables exploring $8\times$ as many solutions with 4 rotations $\times$ 2 flips. We note that additional augmentations and techniques can be performed during inference, which can further boost evaluation performance (Kim et al., 2022; Ma et al., 2022; Choo et al., 2022; Luo

### B.4 EAL Modeling

We describe in more detail the procedure for Efficient Adapter Layers (EAL) modeling. Our initial model trained from Section 5.1 has linear projections layers as referenced in full detail in Appendix B.2 and Appendix B.3 without additional parameters for mixed backhaul and multi-depots.

**EAL for mixed backhauls** This adds, as explained in Section 4.4, a single ($l = 1$) parameter row $\mathbf{W}'_0$ for the mixed backhaul flag $\mu$ to the global embedding. Moreover, we add $l = 1$ rows for the context embedding resulting $\mathbf{W}'_c$ for the available backhaul load $b_t$ at step $t$, i.e. the difference between the vehicle capacity $Q$ and the *used backhaul capacity*.

**EAL for multi-depots** In this case, we do not modify the global embedding but directly project multiple times global attributes and depot locations at each depot node as explained in Appendix B.2. However, we modify the context embedding $\mathbf{W}'_c$ by adding $l = 2$ rows to keep track of the location of the origin depot $o$ we have to return to at step $t$: $[x^o_t, y^o_t]$.

**EAL for multi-depots & mixed backhauls** Here we combine the EAL implementations of the previous two paragraphs. We add the $l = 1$ parameter row $\mathbf{W}'_0$ for the mixed backhaul flag $\mu$ to the global embedding and project the global embedding $m$ according to the number of depots and modify the context embedding $\mathbf{W}'_c$ by adding $l = 3$ rows to keep track of the available backhaul load $b_t$ and the location of the origin depot $[b_t, x^o_t, y^o_t]$.

## C ADDITIONAL MATERIAL

### C.1 DETAILS FOR AVERAGE BATCH REWARD FOR MULTI-TASK REWARD NORMALIZATION

At each training step $t = 1, \ldots, T$ we train on a batch of $b = 1, \ldots, B$ problem instances, each of which belongs to one of the $k \in K$ problem variants covered by ROUTEFINDER. Let $\mathbb{1}_{b,k} \in \{0, 1\}$ be an indicator function such that:

$$\mathbb{1}_{b,k} = \begin{cases} 1 & \text{if instance } b \text{ is of type } k \\ 0 & \text{otherwise} \end{cases}$$

which is efficiently calculated in our unified VRP environment based on vectorized checks. The reward $r^{(k)}_{bt}$ for instance $b$ of variant $k$ at training step $t$ can then be expressed as $r^{(k)}_{bt} = r_{bt} \cdot \mathbb{1}_{b,k}$. The average batch reward $\bar{r}^{(k)}_t$ for variant $k$ at training step $t$ over all instances of type $k$ in a batch can then be expressed as:

$$\bar{r}^{(k)}_t = \frac{\sum_{b=1}^{B} r^{(k)}_{bt}}{\sum_{b=1}^{B} \mathbb{1}_{b,k}} = \frac{\sum_{b=1}^{B} r_{bt} \cdot \mathbb{1}_{b,k}}{\sum_{b=1}^{B} \mathbb{1}_{b,k}}, \qquad \forall k \in K.$$

This average batch reward $\bar{r}^{(k)}_t$ is the basis for the reward normalization explained in Section 4.3.2.

### C.2 HYPERPARAMETER DETAILS

We report in Table C.1 the hyperparameter details common across the main experiments. ROUTEFINDER variants additionally employ the proposed contributions as outlined in the main experiments of Section 5.1.

Table C.1: Experiment hyperparameters. Values with "/" indicate different choices depending on the model, i.e., on the right are values for the Transformer-Based encoder.

| Hyperparameter | Value |
| --- | --- |
| *Model* | |
| Embedding dimension | 128 |
| Number of attention heads | 8 |
| Number of encoder layers | 6 |
| Use Pre-norm | False / True |
| Normalization | Instance / RMSNorm |
| Feedforward hidden dimension | 512 |
| Feedforward structure | MLP / Gated MLP |
| Feedforward activation | ReLU / SwiGLU |
| Tanh clipping | 10.0 |
| Mask logits | True |
| *Training* | |
| Train decode type | multistart sampling |
| Val & Test decode type | multistart greedy |
| Augmentation function | dihedral |
| Batch size | 256 |
| Train data per epoch | 100,000 |
| *Optimization* | |
| Optimizer | Adam |
| Learning rate | 3e-4 |
| Weight decay | 1e-6 |
| LR scheduler | MultiStepLR |
| LR milestones | [270, 295] |
| LR gamma | 0.1 |
| Gradient clip value | 1.0 |
| Max epochs | 300 |

## C.3 Additional Discussion

**Motivation**  Foundation models have been successful in several areas in recent years, including large language models (Achiam et al., 2023), computer vision (Kirillov et al., 2023) as well as other domains such as biology (Abramson et al., 2024; Nguyen et al., 2024). However, foundation models for discrete decision-making, such as CO and our target VRPs, are still under-explored as an area - one reason being the lack of large, high-quality open datasets that can effectively be employed to train such models - which motivates our use of RL. Such foundation models may not only obtain solutions faster than traditional OR counterparts but also avoid the requirement of possibly decades of research and resources to tackle a single task, while a foundation model may automatically learn heuristics without supervision.

**Generalist, or specialized?**  Another open question is the idea of generality behind the model. In ROUTEFINDER, we argue that a model might not need to be extremely complex and be specialized for a specific application (such as routing). One such reason is that with larger model capabilities comes larger size and inference time, which is crucial for real-world deployment. An interesting future direction would be to attempt to generalize a model as a "foundation model for CO", for instance, based on a general formulation (Boisvert et al., 2024), and see whether the additional training and inference costs are worth a (possible) boost in optimality gaps and generalization ability. Such a model may be able to attain a better few-shot generalization to totally unseen attributes, either with adapter layers (Lin et al., 2024) or with our proposed EAL. However, we believe that tailored, specialized foundation models as ROUTEFINDER for VRPs may be more practical and efficient. We note that an orthogonal direction to ours is the use of LLMs as hyper-heuristics (Romera-Paredes et al., 2024; Liu et al., 2024b; Ye et al., 2024a), which starts from a generalist LLM agent to generate algorithms that can be used to improve the optimization of CO problems as VRPs. However, such models are not used at inference time due to the inefficiency of using billions of parameters that are not tailored for the problem at hand.

**Going forward**  in specialized foundation models for VRPs, there are several challenges yet to be addressed. One such challenge is the still sub-par performance compared to state-of-the-art solvers (Wouda & Lan, 2023; Wouda et al., 2024), which may be offset on a larger scale by several means, including decompositions. Another way to attain better performance would be to integrate with local search (Ye et al., 2024b; Kim et al., 2024) and hybridize constructive (the current policy paradigm) with improvement methods (Ma et al., 2021; 2024) to guarantee monotonic improvements given larger time budgets. Finally, given the robust cross-task performance even compared to single-task models, we believe expanding to more VRP variants (and their attribute distributions) may further improve overall performance.

## C.4 Licenses for used assets

Table C.2 lists the used assets and their licenses. Our code is licensed under the MIT License.

Table C.2: Used assets and their licenses.

| Type | Asset | License | Usage |
|------|-------|---------|-------|
| Code | POMO (Kwon et al., 2020) | MIT License | Evaluation |
| | MTPOMO (Liu et al., 2024a) | MIT License | Evaluation |
| | MVMoE (Zhou et al., 2024) | MIT License | Evaluation |
| | RL4CO (Berto et al., 2024) | MIT License | Evaluation |
| | AL (Lin et al., 2024) | MIT License | Evaluation |
| | ORTools (Perron & Didier, 2024) | Apache-2.0 | Evaluation |
| | PyVRP (Wouda et al., 2024) | MIT License | Evaluation |
| Dataset | CVRPLib (Lima et al., 2014) | Available for any non-commercial use | Testing |

## D Additional Empirical Results

This Section supplements the main paper with several experiments evaluating various aspects of ROUTEFINDER:

- Appendix D.1: we study the effect and interactions of Transformer Encoder components.
- Appendix D.2: here we study Mixed Batch Training and its effect on 1) training stability and 2) imbalanced variant distributions.
- Appendix D.3: this section adds additional experiments for zero-shot and finetuning performances with EAL on three unseen new attribute setups: 1) with mixed backhauls 2) with multi-depots and 3) with both mixed backhauls and multi-depots.
- Appendix D.4: here we motivate our ROUTEFINDER foundation model for VRPs when compared to single-variant models in 1) finetuning performance and 2) out-of-distribution generalization.
- Appendix D.5: we evaluation large-scale and real-world distributions in CVRPLIB.
- Appendix D.6: we study the latent learning representation ability of different models via t-SNE across 1) encoding layers 2) effect of different attributes on the latent embeddings.

### D.1 EFFECT OF TRANSFORMER ENCODER COMPONENTS

We study the effect of the proposed Transformer Encoder by ablating its components, in particular:

1. ROUTEFINDER: uses the full proposed Transformer Encoder as described in Section 4.2.1.

2. ROUTEFINDER (No RMSNorm): removes the RMSNorm in pre-norm, but keeps the SwiGLU MLP.

3. ROUTEFINDER (No SwiGLU): removes the SwiGLU MLP, but leaves the RMSNorm

4. ROUTEFINDER (No SwiGLU, No RMSNorm): removes all components and is equivalent to the commonly used Attention Model-style encoder (Kool et al., 2019).

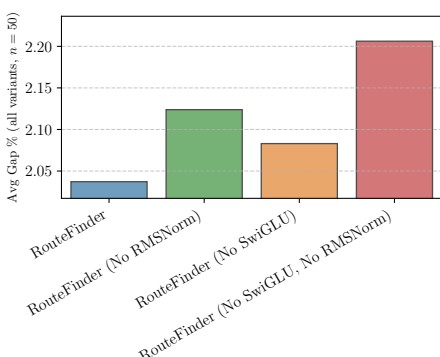

Figure D.1: Effect of encoder components.

We show in Fig. D.1 the effect of each component on the test gaps for $n = 50$ nodes, averaged across the 16 variants of Table D.8. The full ROUTEFINDER provides the best performance. We additionally study the behavior of each single component on validation data during the training epochs across different variants in Appendix D.1. Interestingly, as shown in Appendix D.1, while the final performance for the variant with no RMSNorm outperforms the baseline due to its enhanced capability in representation learning, its convergence is slower in the beginning. However, the full Transformer Encoder containing both RMSNorm and SwiGLU not only performs the best, but also converges the fastest, indicating the importance of each single component.

**FlashAttention speedup** FlashAttention (Dao et al., 2022; Dao, 2023) is a recent exact attention algorithm that can be used to significantly speed up computations with mixed precision. This can be applied to any model with an attention-based mechanism, so we apply it by default to all neural networks compared in this work. Overall, we can improve training and inference speed by up to over 20% with virtually no performance degradation.

### D.2 STUDIES ON MIXED BATCH TRAINING

**Effect on training stability** We visualize the effect of the proposed Mixed Batch Training (MBT) across two different metrics. We compare two ROUTEFINDER models trained with the same hyperparameters on 50 nodes. In Fig. D.3, we show the effect of MBT on the loss function by keeping the overall sampling distribution but mixing variants in the same batch, MBT allows for a much more stable gradient across the different tasks, resulting in a substantially more stable loss compared to training without it. We also show the validation gaps on held-out instances in Fig. D.4, where MBT speeds up convergence across all variants.

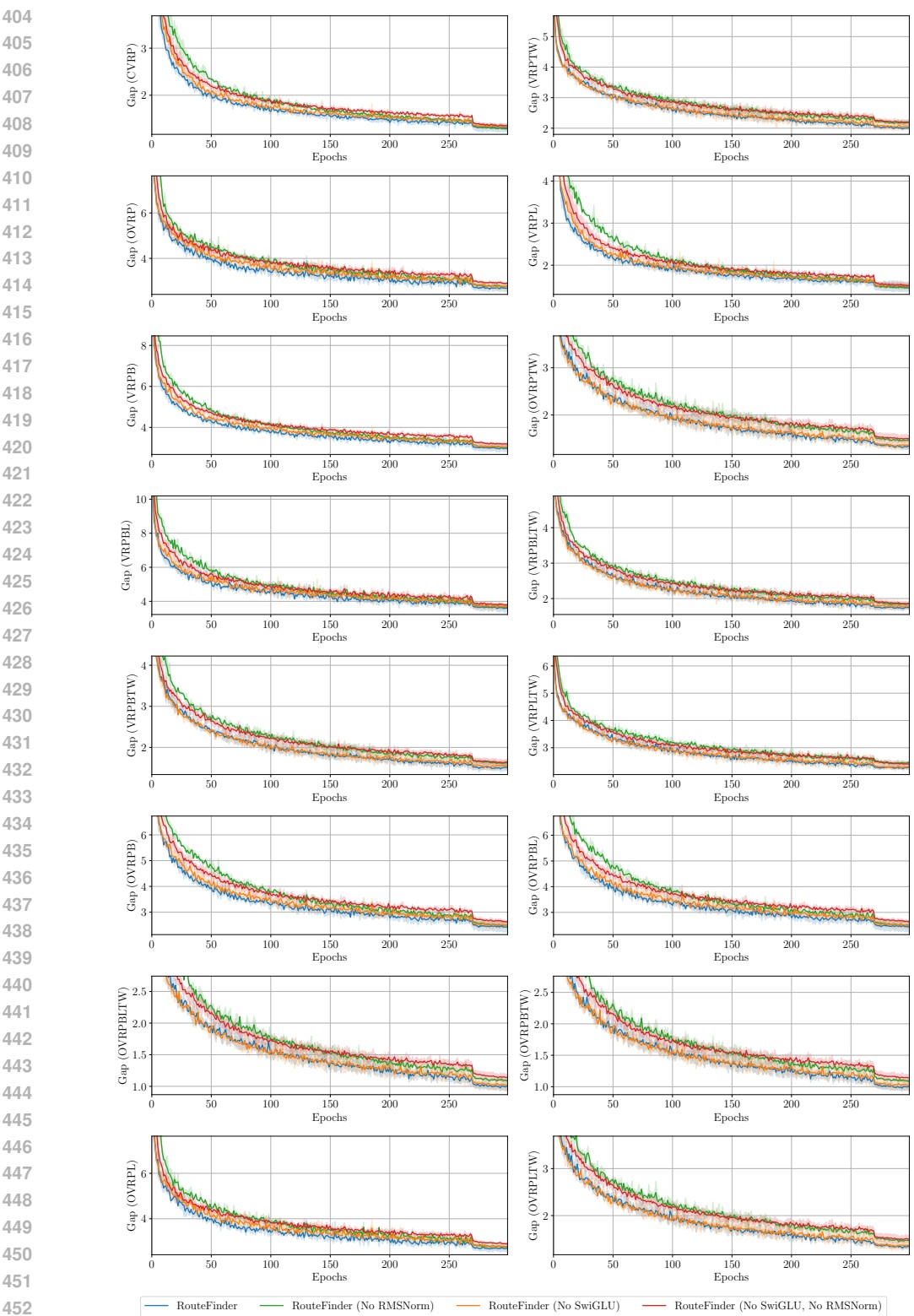

Figure D.2: Ablation study on proposed encoder components over training.

**Effect on imbalanced variant distributions** As explained in Section 4.3.1, we can sample variants uniformly by setting the probability of sampling base attributes $\nu$ as $\mathbf{p}_\nu = 0.5$. We study

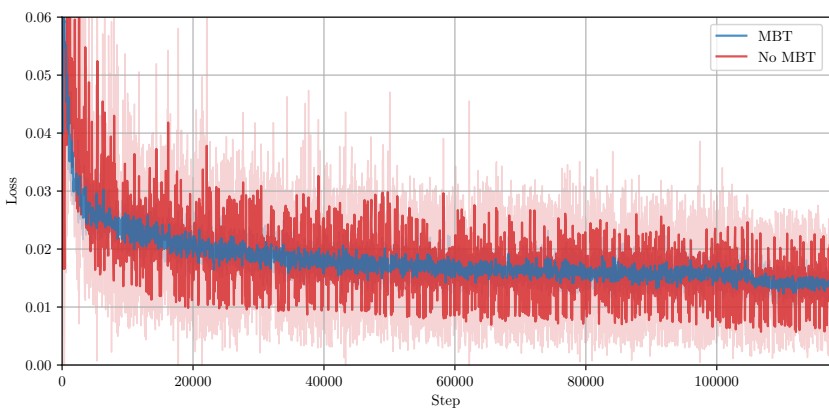

Figure D.3: Stabilizing effect of Mixed Batch Training (MBT) on the loss function on multiple variants.

the behavior of MBT in imbalanced attribute distributions. We train ROUTEFINDER models from scratch with the same setting as the main experiments for 50 epochs with 10,000 instances of size 50 sampled per epoch, with and without MBT, and at different values of the sampling probability for time window attributes $\mathbf{p}_{TW}$ as 0.5, 0.25, and 0.10. Fig. D.5 shows the validation gaps over the training. Decreasing $\mathbf{p}_{TW}$ (towards the right of the plot) results in fewer time window attributes; thus, the convergence is slower for variants such as VRPTW. On the other hand, variants like the CVRP will be sampled with higher probability, which results in slightly faster convergence. MBT plays an important role in stabilizing the training for all cases. Interestingly, while its effect is more moderate for the majority samples (CVRP), this effect is higher on minority samples as VRPTW, where it results in a stable training curve, yielding fast convergence.

### D.3 FINETUNING TO UNSEEN VARIANTS WITH EAL

We conduct additional experiments on zero-shot generalization of various models and finetuning across three different settings of unseen variants in order of difficulty:

1. Mixed backhauls (MB): this is the setting from Section 5.3. We report the results in full in Table D.1 and trends over epochs in Fig. D.6a.

2. Multi-depot (MD): we add additional attribute features for finetuning approaches as per Appendix B.4 with data generated as in Appendix A.1. Results in full are available in Table D.2 and trends over epochs in Fig. D.6b.

3. Mixed backhauls & multi-depot (MB&MD): this is the hardest setting, which considers as finetuning variants only the ones containing both the unseen MB and MB attributes at the same time from Table A.1. Full results are in Table D.3 with trends over epochs in Fig. D.6c.

We keep the same methodology as outlined in Section 5.3, i.e., 10 epochs with 10k instances sampled for each epoch. We use ROUTEFINDER models with Transformer Encoder (RF-TE), untrained for the scratch training and pretrained from the same checkpoints as the main experiments in Section 5.1 for AL and EAL finetuning. Additional details on EAL modeling are available in Appendix B.4.

ROUTEFINDER models perform the best in zero-shot generalization across all experiments; moreover, EAL finetuning achieves the same zero-shot performance as the backbone ROUTEFINDER model RF-TE thanks to the zero-padded initialization, while AL does not due to the introduction of untrained embedding layers. Notably, experiments with multi-depots are much harder than mixed backhaul variants since they require the model to understand multiple starting (and returning) point locations and to schedule vehicle assignments to their respective depots efficiently. EAL performs the best across all variants in finetuning performance. Remarkably, EAL's performance compared to AL and retraining a model from scratch is more prominent with the increasing difficulty of the finetuning task from MB to MB+MD, indicating it is a suitable method for efficient deployment in finetuning to new tasks.

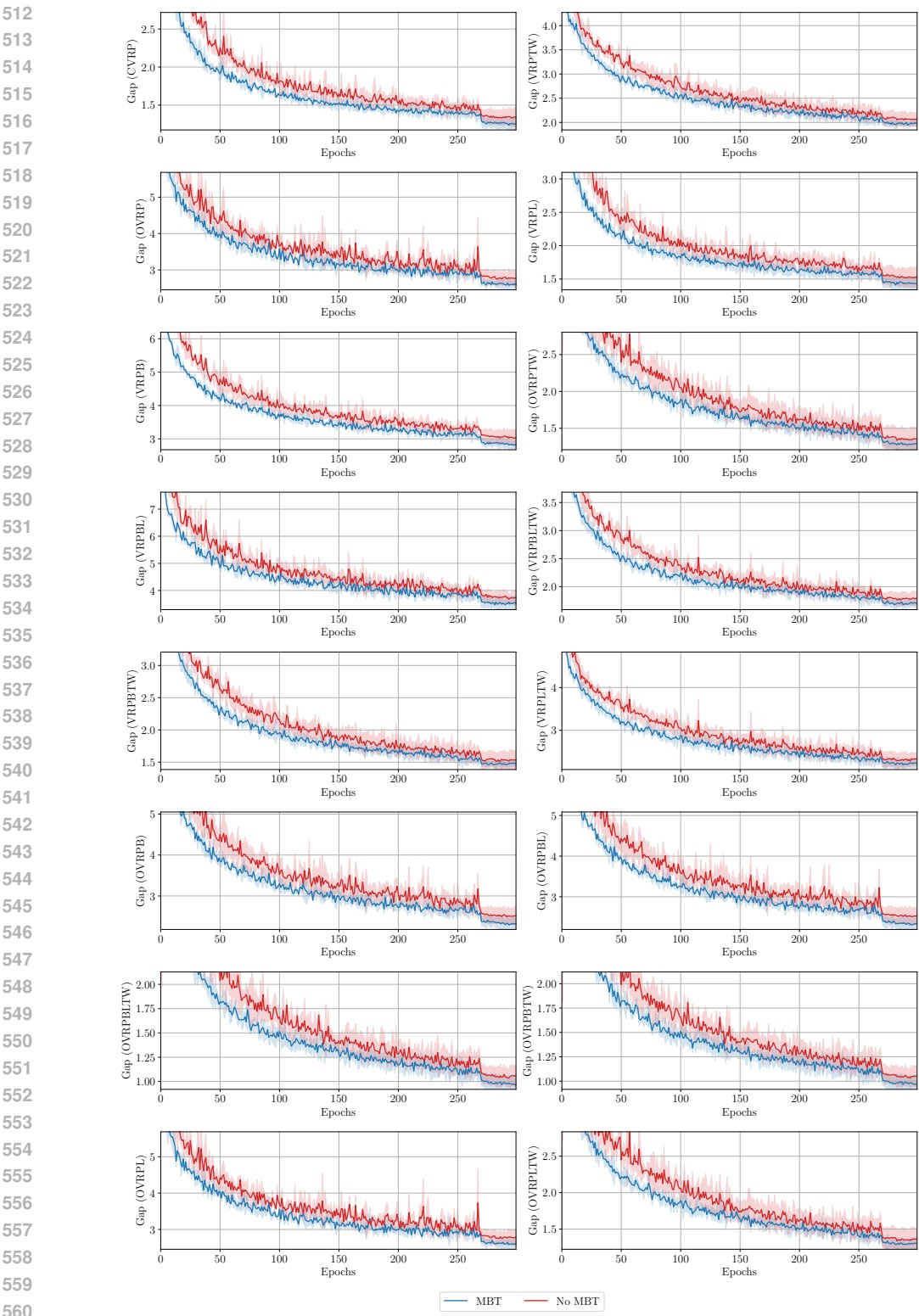

Figure D.4: Mixed Batch Training (MBT) allows for better convergence across all variants.

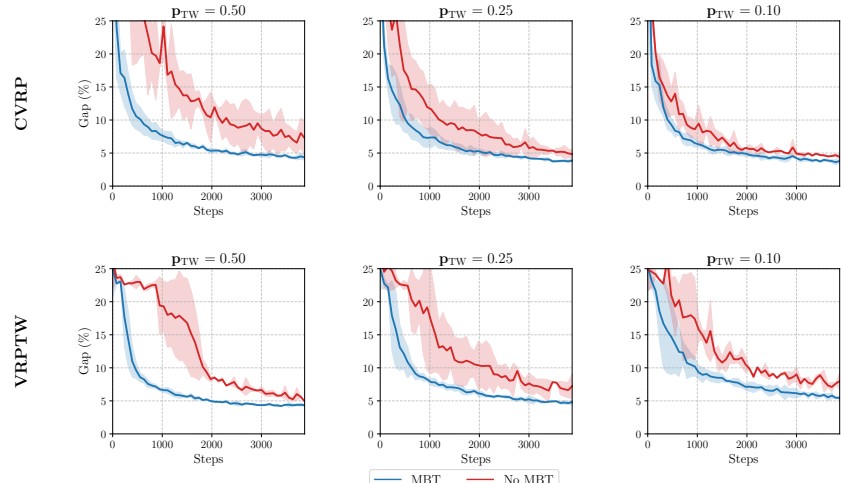

Figure D.5: Effect of Mixed Batch Training (MBT) on imbalanced variant distributions with varying probability $\mathbf{p}_{TW}$ of sampling time windows (TW). MBT stabilizes the training not only for the downsampled TW variants such as VRPTW but also improves the performance for variants with more samples as CVRP.

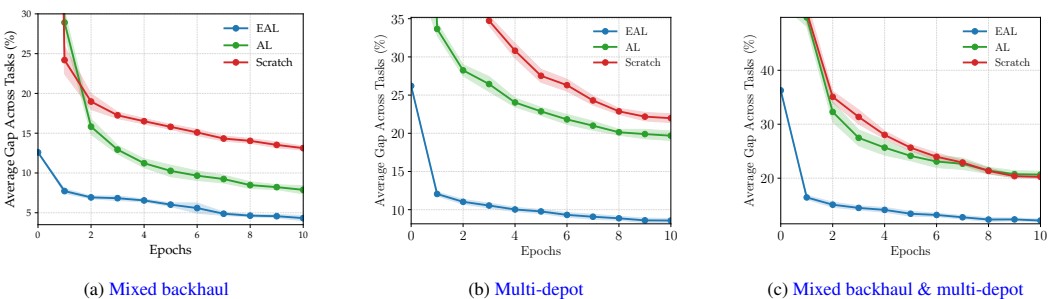

(a) Mixed backhaul   (b) Multi-depot   (c) Mixed backhaul & multi-depot

Figure D.6: Validation gaps averaged across new tasks including unseen features as (a) mixed backhaul (MB), (b) multi-depot (MD), and (c) their combination (MB&MD) for retraining from scratch, AL and EAL finetuning.

Table D.1: Zero-shot, retraining, and fine-tuning performance on unseen mixed backhaul (MB) variants. "$\varnothing$" denotes models and fine-tuning methods evaluated in zero-shot settings. EAL finetuning maintains the zero-shot performance and performs best overall.

| | VRPMB | | OVRPMB | | VRPMBL | | VRPMBTW | | OVRPMBL | | OVRPMBTW | | VRPMBLTW | | OVRPMBLTW | |
|---|---|---|---|---|---|---|---|---|---|---|---|---|---|---|---|---|---|
| Method | Cost | Gap | Cost | Gap | Cost | Gap | Cost | Gap | Cost | Gap | Cost | Gap | Cost | Gap | Cost | Gap |
| HGS-PyVRP | 13.54 | * | 9.01 | * | 13.78 | * | 25.51 | * | 9.01 | * | 16.97 | * | 25.85 | * | 16.97 | * |
| OR-Tools | 14.93 | 10.27% | 10.59 | 17.54% | 15.42 | 11.90% | 29.97 | 17.48% | 10.59 | 17.54% | 19.31 | 13.78% | 30.44 | 17.76% | 19.31 | 13.78% |
| MTPOMO$^{\varnothing}$ | 15.04 | 11.32% | 10.87 | 20.65% | 15.41 | 11.97% | 28.31 | 11.06% | 10.85 | 20.43% | 18.51 | 9.08% | 28.73 | 11.27% | 18.51 | 9.12% |
| MVMoE$^{\varnothing}$ | 14.99 | 10.94% | 10.85 | 20.42% | 15.33 | 11.37% | 28.32 | 11.10% | 10.82 | 20.14% | 18.55 | 9.33% | 28.70 | 11.16% | 18.55 | 9.30% |
| RF-POMO$^{\varnothing}$ | 14.98 | 10.90% | 10.84 | 20.31% | 15.29 | 11.12% | 28.53 | 11.94% | 10.84 | 20.32% | 18.62 | 9.72% | 28.89 | 11.89% | 18.62 | 9.71% |
| RF-MoE$^{\varnothing}$ | 14.93 | 10.49% | 10.76 | 19.49% | 15.21 | 10.47% | 28.20 | 10.63% | 10.76 | 19.40% | 18.45 | 8.74% | 28.55 | 10.57% | 18.45 | 8.72% |
| RF-TE$^{\varnothing}$ | 14.88 | 10.13% | 10.72 | 19.02% | 15.18 | 10.32% | 28.29 | 10.87% | 10.72 | 19.01% | 18.45 | 8.68% | 28.65 | 10.82% | 18.45 | 8.69% |
| Train (scratch) | 15.18 | 12.13% | 10.40 | 15.38% | 15.48 | 12.37% | 28.11 | 10.17% | 10.46 | 16.08% | 18.85 | 11.09% | 28.69 | 10.95% | 18.86 | 11.19% |
| AL$^{\varnothing}$ | 43.15 | 221.25% | 37.98 | 323.23% | 32.81 | 139.84% | 59.17 | 133.55% | 29.15 | 224.37% | 39.03 | 131.09% | 66.62 | 158.21% | 40.92 | 141.51% |
| AL | 14.91 | 10.10% | 10.14 | 12.53% | 15.12 | 9.73% | 27.79 | 8.92% | 10.18 | 12.95% | 18.52 | 9.13% | 28.33 | 9.56% | 18.51 | 9.05% |
| EAL$^{\varnothing}$ | 14.88 | 10.13% | 10.72 | 19.02% | 15.18 | 10.32% | 28.29 | 10.87% | 10.72 | 19.01% | 18.45 | 8.68% | 28.65 | 10.82% | 18.45 | 8.69% |
| EAL | **14.59** | **7.89**% | **9.66** | **7.19**% | **14.78** | **7.39**% | **26.69** | **4.61**% | **9.65** | **7.13**% | **17.60** | **3.70**% | **27.13** | **4.90**% | **17.59** | **3.65**% |

## D.4 COMPARISONS TO SINGLE-VARIANT MODELS

In this section, we study our foundation model and ask the following question: how does ROUTEFINDER perform when compared to models trained specifically on a single variant? To answer this question, we compare ROUTEFINDER and other multi-task learning methods with POMO trained on single variants, including CVRP, VRPL, VRPTW, OVRP, and VRPB. For fairness of

Table D.2: Zero-shot, retraining, and fine-tuning performance on unseen multi-depot (MD) variants. "∅" denotes models and fine-tuning methods evaluated in zero-shot settings. EAL finetuning maintains the zero-shot performance and performs best overall.

| Method | MDCVRP Cost | Gap | MDOVRP Cost | Gap | MDVRPB Cost | Gap | MDVRPL Cost | Gap | MDVRPTW Cost | Gap | MDOVRPTW Cost | Gap | MDOVRPB Cost | Gap | MDOVRPL Cost | Gap |
|---|---|---|---|---|---|---|---|---|---|---|---|---|---|---|---|---|
| HGS-PyVRP | 11.89 | * | 7.97 | * | 11.64 | * | 11.90 | * | 19.33 | * | 13.00 | * | 8.69 | * | 7.97 | * |
| OR-Tools | 12.52 | 5.27% | 8.16 | 2.33% | 12.22 | 5.01% | 12.52 | 5.24% | 19.62 | 1.55% | 13.09 | 0.74% | 8.87 | 2.15% | 8.16 | 2.33% |
| MTPOMO∅ | 16.07 | 35.74% | 10.28 | 29.06% | 15.18 | 30.66% | 16.30 | 37.58% | 26.68 | 38.56% | 17.57 | 35.67% | 10.94 | 26.08% | 10.28 | 29.07% |
| MVMoE∅ | 16.02 | 35.35% | 10.24 | 28.59% | 15.12 | 30.13% | 16.25 | 37.17% | 26.67 | 38.51% | 17.57 | 35.68% | 10.89 | 25.56% | 10.24 | 28.60% |
| RF-POMO∅ | 16.03 | 35.46% | 10.23 | 28.52% | 15.11 | 30.10% | 16.25 | 37.19% | 26.60 | 38.16% | 17.54 | 35.43% | 10.88 | 25.48% | 10.23 | 28.55% |
| RF-MoE∅ | 16.01 | 35.24% | 10.20 | 28.06% | 15.06 | 29.69% | 16.21 | 36.89% | 26.60 | 38.11% | 17.54 | 35.44% | 10.84 | 25.02% | 10.20 | 28.06% |
| RF-TE∅ | 15.98 | 35.02% | 10.18 | 27.82% | 15.05 | 29.53% | 16.20 | 36.76% | 26.51 | 37.64% | 17.48 | 34.96% | 10.82 | 24.74% | 10.18 | 27.84% |
| Train (scratch) | 14.44 | 21.59% | 9.88 | 23.87% | 14.86 | 27.75% | 14.50 | 21.99% | 23.33 | 20.82% | 15.48 | 19.16% | 10.76 | 23.84% | 9.89 | 24.07% |
| AL∅ | 33.91 | 188.76% | 25.02 | 215.12% | 33.56 | 189.58% | 31.06 | 164.78% | 49.08 | 155.57% | 31.17 | 141.42% | 26.30 | 203.65% | 24.12 | 203.73% |
| AL | 14.23 | 19.84% | 9.67 | 21.28% | 14.84 | 27.57% | 14.33 | 20.51% | 22.64 | 17.18% | 15.05 | 15.81% | 10.69 | 23.12% | 9.69 | 21.45% |
| EAL∅ | 15.98 | 35.02% | 10.18 | 27.82% | 15.05 | 29.53% | 16.20 | 36.76% | 26.51 | 37.64% | 17.48 | 34.96% | 10.82 | 24.74% | 10.18 | 27.84% |
| EAL | 12.96 | 9.14% | 8.64 | 8.37% | 13.05 | 12.15% | 12.99 | 9.31% | 21.14 | 9.43% | 13.81 | 6.24% | 9.46 | 8.88% | 8.64 | 8.33% |

| Method | MDVRPBL Cost | Gap | MDVRPBTW Cost | Gap | MDVRPLTW Cost | Gap | MDOVRPBL Cost | Gap | MDOVRPBTW Cost | Gap | MDOVRPLTW Cost | Gap | MDVRPBLTW Cost | Gap | MDOVRPBLTW Cost | Gap |
|---|---|---|---|---|---|---|---|---|---|---|---|---|---|---|---|---|
| HGS-PyVRP | 11.68 | * | 22.03 | * | 19.35 | * | 8.69 | * | 14.369 | * | 13.00 | * | 22.06 | * | 14.37 | * |
| OR-Tools | 12.22 | 4.66% | 22.40 | 1.69% | 19.66 | 1.58% | 8.87 | 2.13% | 14.49 | 0.87% | 13.09 | 0.70% | 22.43 | 1.70% | 14.49 | 0.86% |
| MTPOMO∅ | 15.80 | 35.54% | 30.55 | 39.23% | 27.13 | 40.71% | 10.94 | 26.11% | 19.69 | 37.62% | 17.58 | 35.70% | 31.09 | 41.52% | 19.69 | 37.64% |
| MVMoE∅ | 15.73 | 34.95% | 30.55 | 39.22% | 27.12 | 40.67% | 10.90 | 25.66% | 19.69 | 37.62% | 17.58 | 35.74% | 31.06 | 41.39% | 19.69 | 37.61% |
| RF-POMO∅ | 15.71 | 34.80% | 30.46 | 38.80% | 27.04 | 40.22% | 10.89 | 25.49% | 19.66 | 37.38% | 17.54 | 35.43% | 30.97 | 41.00% | 19.66 | 37.37% |
| RF-MoE∅ | 15.65 | 34.25% | 30.47 | 38.87% | 27.03 | 40.18% | 10.84 | 25.03% | 19.66 | 37.40% | 17.55 | 35.45% | 30.98 | 41.06% | 19.66 | 37.42% |
| RF-TE∅ | 15.62 | 33.98% | 30.36 | 38.36% | 26.93 | 39.69% | 10.82 | 24.78% | 19.59 | 36.95% | 17.48 | 34.95% | 30.86 | 40.49% | 19.60 | 36.96% |
| Train (scratch) | 15.05 | 28.91% | 26.43 | 20.03% | 23.41 | 21.08% | 10.77 | 24.02% | 16.76 | 17.41% | 15.50 | 19.28% | 26.52 | 20.30% | 16.88 | 17.54% |
| AL∅ | 32.08 | 175.25% | 51.70 | 136.04% | 47.65 | 147.90% | 25.00 | 188.85% | 32.45 | 127.08% | 29.94 | 131.91% | 50.14 | 128.59% | 30.93 | 116.41% |
| AL | 14.95 | 28.03% | 25.81 | 17.19% | 22.70 | 17.33% | 10.70 | 23.14% | 16.47 | 14.66% | 15.07 | 15.98% | 25.84 | 17.16% | 16.48 | 14.71% |
| EAL∅ | 15.62 | 33.98% | 30.36 | 38.36% | 26.93 | 39.69% | 10.82 | 24.78% | 19.59 | 36.95% | 17.48 | 34.95% | 30.86 | 40.49% | 19.60 | 36.96% |
| EAL | 13.16 | 12.70% | 23.88 | 8.42% | 21.18 | 9.47% | 9.46 | 8.85% | 15.18 | 5.61% | 13.81 | 6.24% | 23.94 | 8.54% | 15.17 | 5.60% |

Table D.3: Zero-shot, retraining, and fine-tuning performance on unseen variants with combined multi-depots (MD) and mixed backhauls (MB). "∅" denotes models and fine-tuning methods evaluated in zero-shot settings. EAL finetuning maintains the zero-shot performance and performs best overall.

| Method | MDVRPMB Cost | Gap | MDOVRPMB Cost | Gap | MDVRPMBL Cost | Gap | MDVRPMBTW Cost | Gap | MDOVRPMBL Cost | Gap | MDOVRPMBTW Cost | Gap | MDVRPMBLTW Cost | Gap | MDOVRPMBLTW Cost | Gap |
|---|---|---|---|---|---|---|---|---|---|---|---|---|---|---|---|---|
| HGS-PyVRP | 10.68 | * | 7.66 | * | 10.71 | * | 19.29 | * | 7.66 | * | 12.96 | * | 19.31 | * | 12.96 | * |
| OR-Tools | 12.22 | 14.37% | 8.88 | 15.83% | 12.23 | 14.23% | 22.39 | 16.12% | 8.87 | 15.73% | 14.49 | 11.79% | 22.43 | 16.16% | 14.49 | 11.79% |
| MTPOMO∅ | 15.14 | 42.22% | 10.91 | 42.57% | 15.49 | 45.23% | 28.44 | 48.01% | 10.90 | 42.45% | 18.56 | 43.63% | 28.93 | 50.36% | 18.56 | 43.65% |
| MVMoE∅ | 15.08 | 41.67% | 10.90 | 42.41% | 15.40 | 44.37% | 28.46 | 48.12% | 10.88 | 42.13% | 18.61 | 44.04% | 28.89 | 50.19% | 18.60 | 43.95% |
| RF-POMO∅ | 15.09 | 41.78% | 10.90 | 42.41% | 15.37 | 44.05% | 28.68 | 49.27% | 10.90 | 42.37% | 18.69 | 44.70% | 29.08 | 51.15% | 18.69 | 44.69% |
| RF-MoE∅ | 15.02 | 41.08% | 10.82 | 41.40% | 15.29 | 43.34% | 28.38 | 47.67% | 10.82 | 41.36% | 18.50 | 43.19% | 28.77 | 49.56% | 18.50 | 43.22% |
| RF-TE∅ | 14.99 | 40.80% | 10.77 | 40.67% | 15.28 | 43.27% | 28.43 | 47.93% | 10.76 | 40.62% | 18.49 | 43.14% | 28.80 | 49.69% | 18.50 | 43.17% |
| Train (scratch) | 13.12 | 22.88% | 9.37 | 22.32% | 13.24 | 23.72% | 22.85 | 18.56% | 9.38 | 22.44% | 15.13 | 16.75% | 22.90 | 18.65% | 15.11 | 16.60% |
| AL∅ | 34.12 | 223.14% | 26.36 | 245.53% | 27.41 | 158.88% | 48.94 | 155.28% | 24.11 | 216.01% | 31.53 | 144.89% | 46.80 | 143.89% | 30.08 | 133.48% |
| AL | 13.10 | 22.70% | 9.36 | 22.14% | 13.20 | 23.36% | 22.90 | 18.76% | 9.38 | 22.46% | 15.28 | 17.91% | 23.02 | 19.26% | 15.39 | 18.77% |
| EAL∅ | 14.99 | 40.80% | 10.77 | 40.67% | 15.28 | 43.27% | 28.43 | 47.93% | 10.76 | 40.62% | 18.49 | 43.14% | 28.80 | 49.69% | 18.50 | 43.17% |
| EAL | 12.70 | 18.98% | 8.53 | 11.35% | 12.68 | 18.56% | 21.41 | 11.05% | 8.54 | 11.43% | 13.93 | 7.41% | 21.44 | 11.09% | 13.91 | 7.32% |

comparison, we train the POMO models with the same hyperparameters as the other models (from Table C.1), including the same batch size, learning rate, and training epochs on $n = 100$ nodes.

**Finetuning performance** We finetune all POMO models with the same setting as the experiment with unseen mixed backhaul and multi-depots (MB&MD) from Appendix D.3 with EAL.

Table D.4: Fine-tuning performance on unseen variants of single-variant POMO models and ROUTEFINDER. Finetuning a foundation model for VRPs is crucial for fast adaptation to downstream tasks.

| Method | MDVRPMB Cost | Gap | MDOVRPMB Cost | Gap | MDVRPMBL Cost | Gap | MDVRPMBTW Cost | Gap | MDOVRPMBL Cost | Gap | MDOVRPMBTW Cost | Gap | MDVRPMBLTW Cost | Gap | MDOVRPMBLTW Cost | Gap |
|---|---|---|---|---|---|---|---|---|---|---|---|---|---|---|---|---|
| HGS-PyVRP | 10.68 | * | 7.66 | * | 10.71 | * | 19.29 | * | 7.66 | * | 12.96 | * | 19.31 | * | 12.96 | * |
| OR-Tools | 12.22 | 14.37% | 8.88 | 15.83% | 12.23 | 14.23% | 22.39 | 16.12% | 8.87 | 15.73% | 14.49 | 11.79% | 22.43 | 16.16% | 14.49 | 11.79% |
| POMO_CVRP | 13.34 | 24.97% | 9.66 | 26.01% | 13.43 | 25.50% | 25.19 | 30.84% | 9.66 | 25.97% | 25.14 | 30.39% | 17.66 | 36.50% | 17.65 | 36.43% |
| POMO_VRPL | 13.36 | 25.14% | 9.88 | 28.97% | 13.37 | 24.99% | 28.15 | 46.43% | 9.86 | 28.70% | 28.02 | 45.58% | 20.79 | 60.98% | 20.74 | 60.53% |
| POMO_OVRP | 13.31 | 24.62% | 9.54 | 24.45% | 13.35 | 24.77% | 26.03 | 35.27% | 9.55 | 24.63% | 26.03 | 35.07% | 18.65 | 44.21% | 18.66 | 44.30% |
| POMO_VRPTW | 13.91 | 30.27% | 10.17 | 32.77% | 13.99 | 30.72% | 24.70 | 28.13% | 10.22 | 33.43% | 24.78 | 28.43% | 16.74 | 29.32% | 16.80 | 29.77% |
| POMO_VRPB | 13.00 | 21.69% | 9.25 | 20.63% | 13.06 | 22.07% | 22.50 | 16.66% | 9.23 | 20.44% | 22.53 | 16.64% | 14.96 | 15.39% | 14.97 | 15.54% |
| ROUTEFINDER | 12.70 | 18.98% | 8.53 | 11.35% | 12.68 | 18.56% | 21.41 | 11.05% | 8.54 | 11.43% | 13.93 | 7.41% | 21.44 | 11.09% | 13.91 | 7.32% |

Table D.4 shows that fine-tuning our ROUTEFINDER foundation model achieves the best results, even when comparing variants that include only unseen features for both. For instance, POMO trained only on VRP with backhauls (POMO_VRPB in the table) was trained by sampling many

more (classical) backhaul features, but ROUTEFINDER can fine-tune better on MDVRPMB. Models trained on similar features as the target ones, such as POMO_VRPTW, can overall fine-tune better than others on variants that include time windows, as expected, yet not as well as our foundation model. This is a strong motivation for practitioners and researchers: developing foundation models for VRPs is crucial for fast adaptation to new tasks that may arise in real-world scenarios, such as adding new constraints or attributes.

**Out-of-distribution generalization**  We also study out-of-distribution generalization for unseen attribute values of capacities (C), time windows (C), and duration limits (L), for multi-task learning models and single-variant POMO ones. We compare cost values and gaps (the lower, the better) to the results of POMO training specifically for that single variant, similarly to Liu et al. (2024a, Appendix D). All experiments are performed on 1000 variants for each setting with $n = 100$.

For CVRP, the training distribution in 100 nodes considers a vehicle capacity $C = 50$. We study generalization over different capacities $C = \{30, 50, 70, 90, 110, 130, 150, 200\}$ and show the results in Table D.5 with costs. POMO trained specifically on CVRP can perform best for capacities close to the training distribution, while ROUTEFINDER demonstrates a significant improvement for larger capacities.

Table D.5: Comparison of our model with single-task POMO on out-of-distribution CVRP instances.

| Vehicle Capacity | 30 | | 50 | | 70 | | 90 | | 110 | | 130 | | 150 | | 200 | |
|---|---|---|---|---|---|---|---|---|---|---|---|---|---|---|---|---|
| | Cost | Gap | Cost | Gap | Cost | Gap | Cost | Gap | Cost | Gap | Cost | Gap | Cost | Gap | Cost | Gap |
| POMO_CVRP | **22.95** | * | **15.72** | * | **12.91** | * | 11.48 | * | 10.64 | * | 10.04 | * | 9.75 | * | 9.24 | * |
| MTPOMO | 23.29 | 1.50% | 15.87 | 0.94% | 13.07 | 1.24% | 11.69 | 1.77% | 10.88 | 2.30% | 10.34 | 2.90% | 10.04 | 2.97% | 9.59 | 3.77% |
| MVMoE | 23.04 | 0.43% | 15.83 | 0.67% | 12.99 | 0.61% | 11.54 | 0.49% | 10.67 | 0.33% | 10.06 | 0.12% | 9.74 | -0.09% | 9.21 | -0.28% |
| RF-POMO | 23.10 | 0.69% | 15.84 | 0.77% | 13.03 | 0.90% | 11.61 | 1.07% | 10.76 | 1.17% | 10.17 | 1.26% | 9.86 | 1.12% | 9.38 | 1.51% |
| RF-MoE | 23.13 | 0.80% | 15.81 | 0.58% | 13.00 | 0.74% | 11.59 | 0.89% | 10.74 | 0.92% | 10.14 | 0.95% | 9.82 | 0.69% | 9.31 | 0.75% |
| RF-TE | 22.96 | 0.06% | 15.79 | 0.44% | 12.95 | 0.29% | **11.47** | **-0.07%** | **10.56** | **-0.71%** | **9.92** | **-1.22%** | **9.59** | **-1.67%** | **9.02** | **-2.36%** |

In VRPTW, we consider different values of the time interval, i.e., the minimum and maximum values from which service times $s_i$ and time window lengths $t_i$ are sampled (points 1 and 2 for time window generation of Appendix A.1). In distribution, these values are sampled from $[0.15, 0.20]$. In the out-of-distribution settings, we consider them as $\{[0.05, 0.1], [0.15, 0.20], \ldots, [0.85, 0.9], [0.85, 1.0]\}$. The results in Table D.6 demonstrate again that for values differing from the in-training distribution, our model obtains better results than POMO trained solely on VRPTW.

Table D.6: Comparison of our model with single-task POMO on out-of-distribution VRPTW instances.

| Time Interval | [0.05, 0.10] | | [0.15, 0.20] | | [0.25, 0.30] | | [0.35, 0.40] | | [0.45, 0.50] | | [0.55, 0.60] | | [0.65, 0.70] | | [0.75, 0.80] | | [0.80, 0.85] | | [0.95, 1.00] | |
|---|---|---|---|---|---|---|---|---|---|---|---|---|---|---|---|---|---|---|---|---|
| | Cost | Gap | Cost | Gap | Cost | Gap | Cost | Gap | Cost | Gap | Cost | Gap | Cost | Gap | Cost | Gap | Cost | Gap | Cost | Gap |
| POMO_VRPTW | **25.30** | * | **26.27** | * | **28.11** | * | 31.36 | * | 35.25 | * | 39.66 | * | 44.43 | * | 48.17 | * | 52.60 | * | 55.24 | * |
| MTPOMO | 25.51 | 0.84% | 26.59 | 1.20% | 28.27 | 0.57% | 31.28 | -0.26% | 35.05 | -0.56% | 39.51 | -0.39% | 44.34 | -0.21% | 48.25 | 0.17% | 52.85 | 0.47% | 55.67 | 0.78% |
| MVMoE | 25.47 | 0.66% | 26.57 | 1.15% | 28.25 | 0.50% | 31.19 | -0.54% | 34.97 | -0.79% | 39.34 | -0.82% | 44.15 | -0.63% | 48.05 | -0.26% | 52.68 | 0.14% | 55.61 | 0.68% |
| RF-POMO | 25.45 | 0.58% | 26.49 | 0.85% | 28.23 | 0.44% | 31.32 | -0.11% | 35.19 | -0.18% | 39.58 | -0.22% | 44.41 | -0.06% | 48.20 | 0.06% | 52.61 | 0.02% | 55.22 | -0.03% |
| RF-MoE | 25.43 | 0.51% | 26.49 | 0.85% | 28.21 | 0.35% | 31.25 | -0.35% | 35.10 | -0.43% | 39.54 | -0.32% | 44.35 | -0.19% | 48.13 | -0.09% | 52.53 | -0.14% | 55.18 | -0.10% |
| RF-TE | 25.33 | 0.10% | 26.40 | 0.50% | 28.14 | 0.11% | **31.17** | **-0.61%** | **34.91** | **-0.95%** | **39.30** | **-0.93%** | **44.08** | **-0.80%** | **47.86** | **-0.65%** | **52.40** | **-0.38%** | **55.16** | **-0.14%** |

For VRPL, we consider different distance limit values $l$. During training, we sample feasible instances with $l_{max} = 3.0$ as described in Appendix A.1. For out-of-distribution settings, we test distances for values of $l = \{2.9, 3.0, 3.1, 3.2, 3.3, 3.4, 3.5\}$. Interestingly, as shown in Table D.7, our model already outperforms POMO_VRPL in distribution, and the trend is maintained for larger values of $l$.

Finally, Appendix D.5 reports the results for large-scale CVRPLIB, which demonstrate ROUTEFINDER better generalize across sizes and real-world distributions than other multi-task models and single-variant ones. Overall, we can see that ROUTEFINDER is robust, and its advantage is more pronounced the further away from the training distribution we go. This motivates future work in foundation models for VRPs, where we believe that exploring diverse solutions and variants will significantly advance the field.

Table D.7: Comparison of our model with single-task POMO on out-of-distribution VRPL instances.

| Distance Limit | 2.9 | | 3.0 | | 3.1 | | 3.2 | | 3.3 | | 3.4 | | 3.5 | |
|---|---|---|---|---|---|---|---|---|---|---|---|---|---|---|
| | Cost | Gap | Cost | Gap | Cost | Gap | Cost | Gap | Cost | Gap | Cost | Gap | Cost | Gap |
| POMO_VRPL | 15.84 | * | 16.00 | * | 16.04 | * | 15.52 | * | 16.02 | * | 15.74 | * | 15.85 | * |
| MTPOMO | 15.92 | 0.49% | 16.08 | 0.53% | 16.12 | 0.54% | 15.59 | 0.47% | 16.11 | 0.60% | 15.81 | 0.48% | 15.92 | 0.43% |
| MVMoE | 15.88 | 0.22% | 16.03 | 0.22% | 16.08 | 0.27% | 15.54 | 0.11% | 16.04 | 0.15% | 15.78 | 0.25% | 15.88 | 0.20% |
| RF-POMO | 15.91 | 0.41% | 16.04 | 0.27% | 16.09 | 0.33% | 15.56 | 0.28% | 16.06 | 0.29% | 15.78 | 0.29% | 15.87 | 0.13% |
| RF-MoE | 15.86 | 0.12% | 16.03 | 0.21% | 16.05 | 0.08% | 15.53 | 0.09% | 16.04 | 0.15% | 15.77 | 0.21% | 15.87 | 0.12% |
| RF-TE | **15.82** | **-0.17%** | **15.96** | **-0.21%** | **16.02** | **-0.10%** | **15.50** | **-0.10%** | **16.00** | **-0.11%** | **15.72** | **-0.11%** | **15.82** | **-0.16%** |

## D.5 CVRPLIB Evaluation

We report in Table D.8 the results for large-scale CVRPLIB (Lima et al., 2014) with sizes greater than 500 as done in MVMoE (Zhou et al., 2024). We report the original POMO (Kwon et al., 2020) alongside versions of MTPOMO and MVMoE that were initially trained on mixtures of only CVRP, OVRP, VRPL, VRPB, VRPTW, and OVRPTW for more than $3\times$ longer than our setting with all variants. Interestingly, training on all variants improves the generalization performance of MVMoE compared to the original setting, while it decreases the MTPOMO one (possibly due to the fact several more CVRP instances were sampled in MVMoE's setting). Notably, ROUTEFINDER vastly outperforms other SOTA single and multi-task RL baselines.

Table D.8: Results on large-scale CVRPLIB instances from the X set. All models are only trained on the uniformly distributed data with the size $n = 100$ and evaluated via greedy rollouts. Results for methods with † are drawn from Zhou et al. (2024), models trained with single features excluding feature compositions (except for OVRPTW). Training on multiple variants enhances generalization across models.

| Set-X | | POMO[†] | | MTPOMO[†] | | MVMoE[†] | | MVMoE-L[†] | | MTPOMO | | MVMoE | | RF-TE | |
|---|---|---|---|---|---|---|---|---|---|---|---|---|---|---|---|
| Instance | Opt. | Obj. | Gap | Obj. | Gap | Obj. | Gap | Obj. | Gap | Obj. | Gap | Obj. | Gap | Obj. | Gap |
| X-n502-k39 | 69226 | 75617 | 9.232% | 77284 | 11.640% | 73533 | 6.222% | 74429 | 7.516% | 69226 | 9.410% | 76338 | 10.274% | 71791 | 3.705% |
| X-n513-k21 | 24201 | 30518 | 26.102% | 28510 | 17.805% | 32102 | 32.647% | 31231 | 29.048% | 24201 | 42.511% | 32639 | 34.866% | 28465 | 17.619% |
| X-n524-k153 | 154593 | 201877 | 30.586% | 192249 | 24.358% | 186540 | 20.665% | 182392 | 17.982% | 154593 | 14.771% | 170999 | 10.612% | 174381 | 12.800% |
| X-n536-k96 | 94846 | 106073 | 11.837% | 106514 | 12.302% | 109581 | 15.536% | 108543 | 14.441% | 94846 | 16.109% | 105847 | 11.599% | 103272 | 8.884% |
| X-n548-k50 | 86700 | 103093 | 18.908% | 94562 | 9.068% | 95894 | 10.604% | 95917 | 10.631% | 86700 | 27.851% | 104289 | 20.287% | 100956 | 16.443% |
| X-n561-k42 | 42717 | 49370 | 15.575% | 47846 | 12.007% | 56008 | 31.114% | 51810 | 21.287% | 42717 | 30.770% | 53383 | 24.969% | 49454 | 15.771% |
| X-n573-k30 | 50673 | 83545 | 64.871% | 60913 | 20.208% | 59473 | 17.366% | 57042 | 12.569% | 50673 | 20.210% | 61524 | 21.414% | 55952 | 10.418% |
| X-n586-k159 | 190316 | 229887 | 20.792% | 208893 | 9.761% | 215668 | 13.321% | 214577 | 12.748% | 190316 | 19.125% | 212151 | 11.473% | 205575 | 8.018% |
| X-n599-k92 | 108451 | 150572 | 38.839% | 120333 | 10.956% | 128949 | 18.901% | 125279 | 15.517% | 108451 | 21.098% | 126578 | 16.714% | 116560 | 7.477% |
| X-n613-k62 | 59535 | 68451 | 14.976% | 67984 | 14.192% | 82586 | 38.718% | 74945 | 25.884% | 59535 | 30.523% | 73456 | 23.383% | 67267 | 12.987% |
| X-n627-k43 | 62164 | 84434 | 35.825% | 73060 | 17.528% | 70987 | 14.193% | 70905 | 14.061% | 62164 | 23.193% | 70414 | 13.271% | 67572 | 8.700% |
| X-n641-k35 | 63682 | 75573 | 18.672% | 72643 | 14.071% | 75329 | 18.289% | 72655 | 14.090% | 63682 | 30.321% | 71975 | 13.023% | 70831 | 11.226% |
| X-n655-k131 | 106780 | 127211 | 19.134% | 116988 | 9.560% | 117678 | 10.206% | 118475 | 10.952% | 106780 | 12.731% | 119057 | 11.497% | 112202 | 5.078% |
| X-n670-k130 | 146332 | 208079 | 42.197% | 190118 | 29.922% | 197695 | 35.100% | 183447 | 25.364% | 146332 | 24.809% | 168226 | 14.962% | 168999 | 15.490% |
| X-n685-k75 | 68205 | 79482 | 16.534% | 80892 | 18.601% | 97388 | 42.787% | 89441 | 31.136% | 68205 | 36.550% | 82269 | 20.620% | 77847 | 14.137% |
| X-n701-k44 | 81923 | 97843 | 19.433% | 92075 | 12.392% | 98469 | 20.197% | 94924 | 15.870% | 81923 | 13.319% | 90189 | 10.090% | 89932 | 9.776% |
| X-n716-k35 | 43373 | 51381 | 18.463% | 52709 | 21.525% | 56773 | 30.895% | 52305 | 20.593% | 43373 | 37.657% | 52250 | 20.467% | 49669 | 14.516% |
| X-n733-k159 | 136187 | 159098 | 16.823% | 161961 | 18.925% | 178322 | 30.939% | 167477 | 22.976% | 136187 | 28.910% | 156387 | 14.833% | 148463 | 9.014% |
| X-n749-k98 | 77269 | 87786 | 13.611% | 90582 | 17.229% | 100438 | 29.985% | 94497 | 22.296% | 77269 | 32.182% | 92147 | 19.255% | 85171 | 10.227% |
| X-n766-k71 | 114417 | 135464 | 18.395% | 144041 | 25.891% | 152352 | 33.155% | 136255 | 19.086% | 114417 | 16.692% | 130505 | 14.061% | 129935 | 13.563% |
| X-n783-k48 | 72386 | 90289 | 24.733% | 83169 | 14.897% | 100383 | 38.677% | 92960 | 28.423% | 72386 | 50.140% | 96336 | 33.087% | 83185 | 14.919% |
| X-n801-k40 | 73305 | 124278 | 69.536% | 85077 | 16.059% | 91560 | 24.903% | 87662 | 19.585% | 73305 | 24.536% | 87118 | 18.843% | 86164 | 17.542% |
| X-n819-k171 | 158121 | 193451 | 22.344% | 177157 | 12.039% | 183599 | 16.113% | 185832 | 17.525% | 158121 | 22.148% | 179596 | 13.581% | 174441 | 10.321% |
| X-n837-k142 | 193737 | 237884 | 22.787% | 214207 | 10.566% | 229526 | 18.473% | 221286 | 14.220% | 193737 | 19.429% | 230362 | 18.904% | 208528 | 7.635% |
| X-n856-k95 | 88965 | 152528 | 71.447% | 101774 | 14.398% | 99129 | 11.425% | 106816 | 20.065% | 88965 | 33.103% | 105801 | 18.924% | 98291 | 10.483% |
| X-n876-k59 | 99299 | 119764 | 20.609% | 116617 | 17.440% | 119619 | 20.463% | 114333 | 15.140% | 99299 | 15.240% | 114016 | 14.821% | 107416 | 8.174% |
| X-n895-k37 | 53860 | 70245 | 30.421% | 65587 | 21.773% | 79018 | 46.710% | 64310 | 19.402% | 53860 | 96.818% | 69099 | 28.294% | 64871 | 20.444% |
| X-n916-k207 | 329179 | 399372 | 21.324% | 361719 | 9.885% | 383681 | 16.557% | 374016 | 13.621% | 329179 | 18.134% | 373600 | 13.494% | 352998 | 7.236% |
| X-n936-k151 | 132715 | 237625 | 79.049% | 186262 | 40.347% | 220926 | 66.466% | 190407 | 43.471% | 132715 | 50.654% | 161343 | 21.571% | 163162 | 22.942% |
| X-n957-k87 | 85465 | 130850 | 53.104% | 98198 | 14.898% | 113882 | 33.250% | 105629 | 23.593% | 85465 | 48.127% | 123633 | 44.659% | 102689 | 20.153% |
| X-n979-k58 | 118976 | 147687 | 24.132% | 138092 | 16.067% | 146347 | 23.005% | 139682 | 17.404% | 118976 | 16.711% | 131754 | 10.740% | 129952 | 9.225% |
| X-n1001-k43 | 72355 | 100399 | 38.759% | 87660 | 21.153% | 114448 | 58.176% | 94734 | 30.929% | 72355 | 82.677% | 88969 | 22.962% | 85929 | 18.760% |
| Avg. Gap | | | 29.658% | | 16.796% | | 26.408% | | 19.607% | | 30.202% | | 18.795% | | **12.303%** |

## D.6 T-SNE VISUALIZATION

For interpretability, we study the representations learned from the model across different variants. Given their high dimensionality, we employ t-SNE (Van der Maaten & Hinton, 2008) to project them in 2D space. We employ the implementation from `scikit-learn` with the default perplexity of 30 and use 100 instances of size 100 for each of the 16 variants of the main experiments from Section 5.1.

**Layer-wise visualization** We study ROUTEFINDER's Transformer Encoder layers. As shown in Fig. D.7, distinct clusters emerge at different model layers, indicating that the model progressively separates the problem variants with increasing depth. Early layers (Layer 1) exhibit high overlap between different variants, suggesting shared feature extraction. However, as we proceed to deeper layers (Layer 6), the clusters become more distinct, particularly for more complex variants such as OVRPB, VRPBLTW, and VRPBTW, signifying the model's capacity to capture and differentiate intricate problem structures.

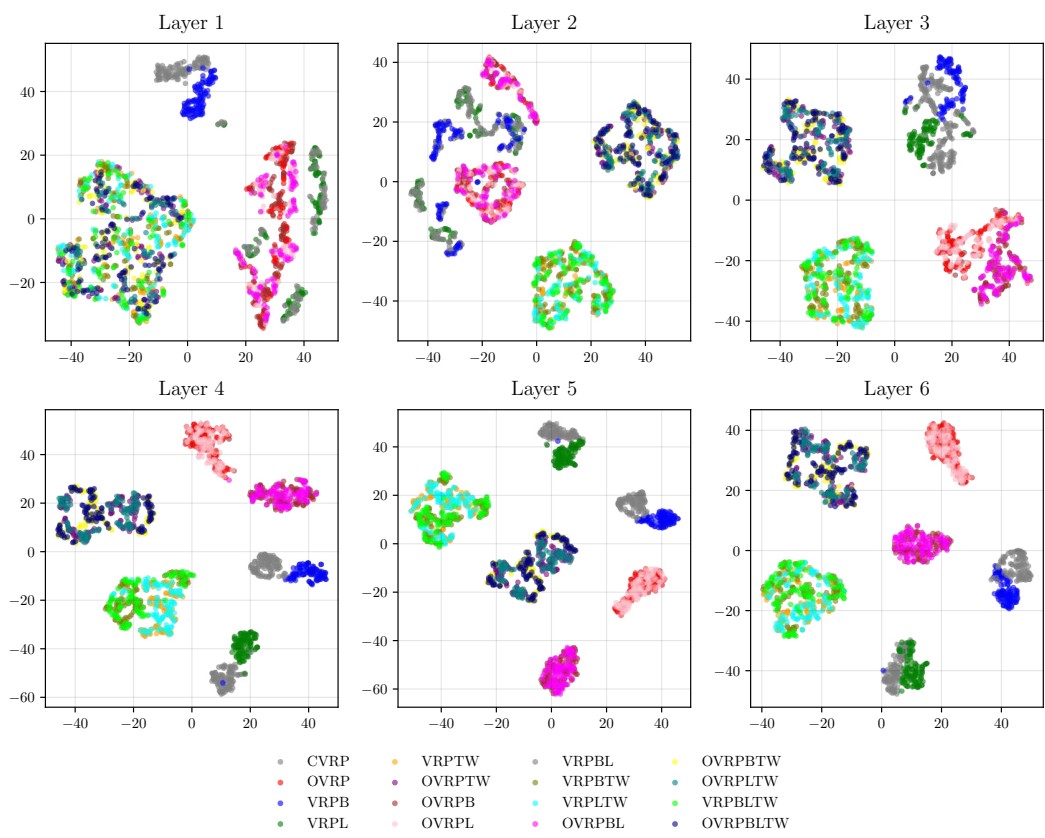

Figure D.7: Visualization of ROUTEFINDER's Transformer Encoder latent space via t-SNE analysis by layer. Problem patterns become more visible with deeper layers, generating distinct clusters.

**Comparison across models and VRP variants** We also compare t-SNE analyses across the models, in particular, MTPOMO and MVMoE, compared to our ROUTEFINDER with Transformer Encoder layers, with embeddings taken in the last encoder layer for all models. In particular, we aim to analyze the differences in latent representation problem variants across the four attributes: open routes (O), distance limits (L), backhauls (B), and time windows (TW). Fig. D.8 shows that ROUTEFINDER generates more and defined clusters, indicating a better-learned representation (Arora et al., 2018). For open routes, ROUTEFINDER has more defined clusters than the baselines. In distance limits, our model generates double the clusters, which indicates different relations between attributes; for instance, the model clearly separates backhaul variants VRPB and VRPBL (green and grey, respectively), while other models do not clearly do this. This also holds in the backhaul

attribute clusters, where ROUTEFINDER more clearly separates different types of time windows as well as distance limits. Finally, for time windows clusters, we notice the most striking difference – while MTPOMO and MVMoE fail to distinguish between time window variants, resulting in a single and sparse cluster, ROUTEFINDER separates time window variants with and without the open (O) attribute into two separate clusters thanks to the Global Attribute Embeddings.

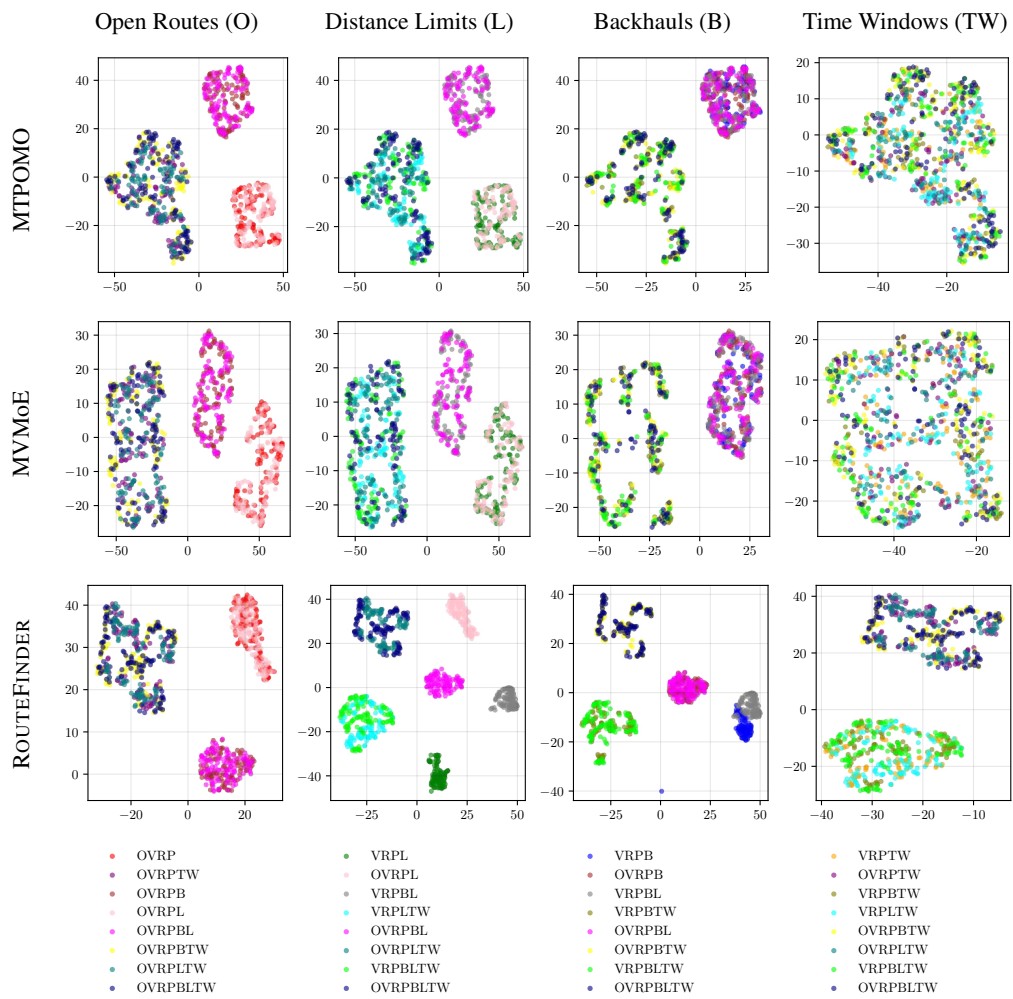

Figure D.8: Analysis of the t-SNE latent space for the last encoder layer across different attributes. ROUTEFINDER yields well-defined, tightly grouped, and distinct clusters on all variants, which is a strong indicator of its capability to generalize and specialize effectively in solving diverse VRP variants. For example, unlike baselines, ROUTEFINDER distinctly separates time window variants into two clusters with and without open routes (bottom-right image) thanks to the Global Attribute Embeddings.

