# OpenReview forum: "RouteFinder: Towards Foundation Models for Vehicle Routing Problems"
_ICLR.cc/2025/Conference — Submitted to ICLR 2025_

### Official Review · Reviewer_p8aH · 2024-11-01

**Soundness:** 2
**Presentation:** 3
**Contribution:** 2
**Rating:** 6
**Confidence:** 2

**Summary:**

This paper introduces RouteFinder, a foundation model for multiple VRP variants, using a transformer-based encoder with Global Attribute Embeddings. Key innovations include Mixed Batch Training (MBT), Multi-Variant Reward Normalization, and Efficient Adapter Layers (EAL) for adaptability to new problem variants. While it shows competitive results across 24 VRP variants, some limitations remain in performance on certain benchmarks.

**Strengths:**

1.RouteFinder is a significant step toward a general model for VRP, reducing the need for separate models for each variant and improving scalability.

2.The use of Global Attribute Embeddings and a transformer-based encoder helps the model differentiate between VRP variants and adapt effectively.

3.Mixed Batch Training (MBT) improves stability by training on multiple variants at once.

**Weaknesses:**

1.Despite generalization, RouteFinder lags behind specialized models on certain VRP variants, such as LKH, HGS, or SISR. A deeper analysis of these trade-offs is needed.

2.The limitations of MBT in highly imbalanced datasets are not well explored. More ablation studies would be helpful to understand its optimal conditions.

3.There is a lack of discussion on how the different model components interact. Further exploration of these interactions would be valuable.

**Questions:**

1.How does the difference between AL and EAL affect the outcome, given that EAL, by maintaining a predefined zero matrix, doesn't seem to have an impact on AL, which later fills in the matrix during training? It would be helpful to include empirical results or visualizations that demonstrate the effects of these two approaches at different stages of training.

2.Could you discuss the potential trade-offs between the two methods and scenarios where one might be more advantageous than the other?

---

> ### Author Response · Authors · 2024-11-13
>
> Dear Reviewer p8aH,
>
> We noticed that the review you provided seems to be for a different paper. Could you kindly share your review content specific to our submission?
>
> Thank you very much for your time and effort in helping us improve our work.
>
> Best regards,
> RouteFinder Authors

---

> ### Author Response · Authors · 2024-11-25
>
> Thank you for providing the updated review. We will provide detailed responses, where `W` denotes weaknesses and `Q` denotes questions.
>
> **W1. Performance trade-offs when compared to models designed for specific problem variants are not studied.**
>
> Indeed, training a model on a single variant may yield better results in-distribution. However, RouteFinder is much more robust than models trained on single variants when evaluated on out-of-distribution cases. For instance, in large-scale CVRPLIB (Table D.8 in Appendix), RouteFinder is much better than POMO trained specifically for CVRP. We added new experiments in Section D.4, where we compare the single-variant POMO trained on specific variants such as CVRP, VRPL and VRPTW and compare the results with RouteFinder on out-of-distribution cases for capacities, time windows, and distance limits. Interestingly, we find that while POMO trained on specific variants, as expected, can perform better close to the training distribution, it performs worse out-of-distribution, whereas RouteFinder demonstrates exceptional generalization.
>
>
> **W2. Missing analysis on MBT limitations for significantly imbalanced distributions.**
>
> To address this, we ran new experiments to analyze MBT's performance under significantly imbalanced attribute distributions available in Section D.2 of the Appendix. Using our RouteFinder model with and without MBT, we ran training runs at different sampling probabilities of time-window attributes $p_\text{tw} = \{  0.5, 0.25, 0.10\}$ to examine different levels of imbalance.  The results (Figure D.5) show that as $p_\text{tw}$ decreases, convergence slows for challenging variants like VRPTW, while majority-dominant variants (e.g., CVRP) converge faster. Crucially, MBT stabilizes training across all settings, particularly benefiting minority variants by mitigating instability and ensuring smoother, faster convergence. This demonstrates that MBT is a robust strategy for improving training stability and convergence in imbalanced scenarios.
>
>
> **W3. Ablation studies do not show the interaction among components.**
>
> Figure 5.1 shows the ablation studies of several RouteFinder components alongside a sensitivity analysis on Reward Normalization. All the proposed components significantly improve the performance of our model. We also added additional ablation studies, especially regarding the Transformer Encoder components in Appendix D.1. Here, we compare RouteFinder with no SwiGLU and no RMSNorm, with SwiGLU only, with RMSNorm only, and with both (full Transformer Encoder). The results demonstrate that the Transformer Encoder performs best.
>
>
> **Q1. How do AL and EAL differ?**
>
> EAL differs from AL because it keeps the first initial $k$ linear projection rows the same, hence inheriting them from the pretrained model. To ensure compatibility with a new task with $k+l$ raw features (with $l$ new features) we add $l$ new rows initialized with zeros. This has an important property. Suppose we trained the model on a multitude of variants, including CVRP. If we use AL, we lose the information of the pretrained linear projections for CVRP, hence the model will not be able to construct CVRP solutions unless finetuning is applied. On the contrary, EAL keeps this information, which can be _transferred_ for new tasks - for example, for a multi-depot VRP, we would like to keep the learned information about depot locations, since they are important for VRPs.
>
> This means that EAL, in practice, can keep the zero-shot performance of a pretrained model when applied to a new task. We added more evidence for this phenomenon throughout new experiments in Tables D1 to D3, and also show the importance of EAL in new Figure D.6 across epochs where it can be seen how EAL yields fast finetuning to new variants.
>
> **Q2. What are the potential trade-offs between AL and EAL?**
>
> AL might be preferable to EAL only if the new tasks do not share any raw feature with the previously seen tasks. However, in the context of VRPs, this would rarely happen since features like locations and demands are required in the very definition of VRPs. Thus, in Foundation Models for VRPs, EAL finetuning would almost always be preferable to AL.

---

> > ### Author Response · Authors · 2024-12-02
> > **Request for Feedback**
> >
> > As the author-reviewer discussion phase is nearing its conclusion (less than 24 hours remaining), we kindly ask for your feedback on our responses and the extensive revisions made to our paper ([view revised version](https://openreview.net/pdf?id=du9reSRIo1)). If our revisions adequately address your concerns, we would greatly appreciate it if you could reconsider your evaluation.
> >
> > Should any additional questions or concerns arise, we are more than happy to provide further clarifications. Thank you for your time and thoughtful consideration.

---

### Official Review · Reviewer_kTqa · 2024-11-03

**Soundness:** 2
**Presentation:** 3
**Contribution:** 2
**Rating:** 5
**Confidence:** 3

**Summary:**

The paper introduces ROUTEFINDER, a foundation model designed to tackle various Vehicle Routing Problem (VRP) variants through a unified framework. It presents several key components, including a Transformer-based model architecture, Global Attribute Embeddings, Mixed Batch Training, and Efficient Adapter Layers for fine-tuning unseen attributes. While the paper proposes an interesting approach and demonstrates competitive experimental results across multiple VRP variants, there are concerns regarding the depth of innovation in the proposed methods and the clarity of explanations in some sections.

**Strengths:**

1. The authors propose a unified VRP environment that can handle multiple VRP variants efficiently, demonstrating a strong attempt at generalization in a complex combinatorial optimization domain.
2. The introduction of Efficient Adapter Layers (EAL) presents a lightweight yet powerful method to fine-tune the model for unseen attributes, which is crucial for practical applications that require adaptability.

**Weaknesses:**

1. Figure 4.1 lacks sufficient explanation. There is a need for a more detailed description of Figure 4.1.
2. The proposed unified VRP environment and attribute dynamic composition approach, while having some differences compared to MTPOMO and MVMoE in handling different attributes, does not present substantial changes. In fact, MTPOMO and MVMoE can also achieve similar functionality.
3. The improvements to the Transformer architecture are mainly module-level substitutions, such as the use of RMS Normalization, SwiGLU activation, and FlashAttention. While these changes enhance training stability and convergence, they largely involve the application of existing techniques rather than introducing significant innovation.
4. Lack of evaluation on large-scale.

**Questions:**

1. The design of Global Attribute Embeddings, such as Open and Limit attributes, is a notable feature. However, in MTPOMO and MVMoE, these attributes are already considered in decoder. What is the significance of redundantly including these embeddings in the encoder as well in this work?
2. The meaning of $l_l$ is unclear. Does $l_{rand(0,1)}<1/2$ represent random sampling of half of the attributes?
3. The explanation of why encountering different problem variants in a single batch leads to more stable training (as shown in Figure 4.3) is not well substantiated. Additional evidence is needed to support this claim.
4. The computation of the average reward $\bar{r}^{(k)}_t$ needs to be explained in more detail.

---

> ### Author Response · Authors · 2024-11-25
>
> We thank you for your effort and for providing constructive feedback in reviewing our paper! We will provide detailed responses, where `W` denotes weaknesses and `Q` denotes questions.
>
> **W1. Insufficient explanation for Figure 4.1.**
>
> We agree that our previous caption was very brief and potentially insufficient. We provided a better explanation in the revised paper that we report here: *"Overview of RouteFinder. The unified VRP environment is used for generating data and performing rollouts (Section 4.1). Our Transformer-based encoder (Section 4.2.1) is employed to process node and global embeddings (Section 4.2.2) of problem instances. During training, we sample multiple variants in the same batch (Section 4.3.1) whose multi-task reward is then normalized (Section 4.3.2). Efficient Adapter Layers (EAL) can be employed for efficient fine-tuning to new variants (Section 4.4)"*.
>
>
> **W2. RouteFinder does not present substantial changes.**
>
> RouteFinder is an end-to-end approach that greatly enhances existing models. Our contributions go beyond a single change; they span the entire pipeline, from the environment setup to fine-tuning, and include several novel components. In particular: 1) Environment: new unified VRP environment 2) Model: new Transformer Encoder and Global Attribute Embeddings 3) Training: new Mixed Batch Training and Multi-Variant Reward Normalization 4) new Efficient Adapter Layers finetuning method for VRPs. During the rebuttal, we also added several new experiments, including doubling the number of variants modeled by the unified VRP environment: as such, our RouteFinder is the first neural method to the best of our knowledge to model 48 VRP variants, while previous approaches as MTPOMO and MVMoE modeled 16. Finally, we remark that our contributions achieve SOTA in learning foundation models for VRPs.
>
> **W3. The improvements to the transformer architecture employ existing techniques.**
>
> The Transformer Encoder layers improvements are inspired by recent research in LLMs. In this sense, the single components are not our innovation. However, we would like to note that RouteFinder is the first model, to the best of our knowledge, to bring such components to learned methods for VRPs — that mostly still use architectures introduced several years ago — and we believe our adoption will advance this field. Moreover, we studied the combination of these components in new experimental results, which are available in Appendix D.1. Single components such as the SwiGLU MLP might actually degrade performance initially, as shown in Figure D.2, if not coupled with, e.g., the RMSNorm. Their combination reveals synergistic behavior, which greatly improves baseline encoder performance and motivates the development of new Transformer Encoder layers.
>
> **W4. Lack of evaluation on large-scale.**
>
> We studied the large-scale real-world CVRPLIB with up to 1000 nodes in Appendix D.5. Notably, RouteFinder not only outperforms other SOTA multi-task RL baselines but also single-task RL solvers dedicated to the CVRP, underpinning the importance of training on a diverse set of tasks for generalizing to larger scales.

---

> > ### Author Response · Authors · 2024-11-25
> >
> > **Q1. Why should we include global attributes in the encoder, since MTPOMO and MVMoE already do so in the decoder?**
> >
> > This is because the decoder is not deep and thus cannot easily distinguish the differences between tasks easily; considering these global attributes only in the decoder does not allow for good representation. Indeed, the encoder holds the majority of the parameters and is essential for it to process the data for learning representations through deep layers.
> >
> > To better motivate our addition of Global Attribute Embeddings, we added new experiments visualizing the hidden space via t-SNE between transformer layers across all models to understand how RouteFinder and previous models understand different attributes (Figures D.7 and D.8 in Appendix). We note that RouteFinder exhibits more complex patterns for deeper layers. Interestingly, when analyzing the hidden space on a _per attribute_ basis, our model yields well-defined, tightly grouped, and distinct clusters on all variants, which is a strong indicator of its capability to generalize and specialize effectively in solving diverse VRP variants. For example, unlike baselines, RouteFinder distinctly separates time window variants into two clusters with and without open routes thanks to the Global Attribute Embeddings.
> >
> > **Q2. Does \\( \\mathbf{1}\_{\\text{rand}(0,1)} < 1/2 \\) represent random sampling of half of the attributes?**
> >
> > That is correct. We will clarify this a little: \\( \\mathbf{1}\_{\\text{rand}(0,1)} < \\mathbf{p}\_{\\nu} \\) represents the probability of sampling variants with a specific attribute \\( \\nu \\). If this \\( \\mathbf{p}\_{\\nu} = 1/2 \\), then this indeed means we have a 50\\% chance of sampling that specific attribute. For example, we added new experiments on imbalanced attribute data in Appendix D.2 in which the probability of sampling time windows (TW) variants \\( \\mathbf{p}\_{TW} < 0.5 \\). In such cases, as we can see in the new Figure D.5, we can see that sampling less time windows, such as only 10\\% of sampled data contains time windows, has an effect on slower convergence for TW variants. However, this issue is successfully mitigated with the help of our proposed MBT.
> >
> > **Q3. Can you provide additional evidence for why different problem variants in a single batch leads to more stable training?**
> >
> > Mixed Batch Training (MBT) provides a smoother loss function landscape that helps optimization, as shown in Figure 4.3. The ablation study in Section 5.3 also provides evidence on its effect. We additionally provide new experiments to substantiate our claim which we believe provide better insights in Appendix D.2. In particular, Figure D.3 shows the effect of MBT on the loss function by keeping the overall sampling distribution but mixing variants in the same batch, MBT allows for a much more stable gradient across the different tasks, resulting in a substantially more stable loss compared to training without it. We also show the validation gaps on held-out instances in Fig. D.4, where MBT speeds up convergence across all variants. We finally show this holds true for imbalanced attribute distributions as well in Figure D.5.
> >
> > **Q4. Could you explain more in detail the computation of the average reward?**
> >
> > We will provide the detailed explanation of the average reward as follows.
> >
> > At each training step \\( t = 1, \\dots, T \\), we train on a batch of \\( b = 1, \\dots, B \\) problem instances, each of which belongs to one of the \\( k \\in K \\) problem variants covered by RouteFinder.
> >
> > Let \\( \\mathbf{1}\_{b,k} \\in \\{0,1\\} \\) be an indicator function such that:
> >
> > \\[
> > \\mathbf{1}\_{b,k} =
> > \\begin{cases}
> >     1 & \\text{if instance } b \\text{ is of type } k \\\\
> >     0 & \\text{otherwise}
> > \\end{cases}
> > \\]
> >
> > This is efficiently calculated in our unified VRP environment based on vectorized checks.
> >
> > The reward \\( r\_{bt}^{(k)} \\) for instance \\( b \\) of variant \\( k \\) at training step \\( t \\) can then be expressed as:
> >
> > \\[
> > r\_{bt}^{(k)} = r\_{bt} \\cdot \\mathbf{1}\_{b,k}.
> > \\]
> >
> > The average batch reward \\( \\bar{r}\_t^{(k)} \\) for variant \\( k \\) at training step \\( t \\) over all instances of type \\( k \\) in a batch can then be expressed as:
> >
> > \\[
> > \\bar{r}\_t^{(k)}
> > = \\frac{\\sum\_{b=1}^{B} r\_{bt}^{(k)}}{\\sum\_{b=1}^{B} \\mathbf{1}\_{b,k}}
> > = \\frac{\\sum\_{b=1}^{B} r\_{bt} \\cdot \\mathbf{1}\_{b,k}}{\\sum\_{b=1}^{B} \\mathbf{1}\_{b,k}} ,
> > \\qquad \\forall k \\in K.
> > \\]
> >
> > This average batch reward \\( \\bar{r}\_t^{(k)} \\) serves as the basis for the reward normalization explained in Section 4.3.2.
> >
> > We have added the above in Appendix C.1 for clarity.

---

> > > ### Author Response · Authors · 2024-12-02
> > > **Request for Feedback**
> > >
> > > As the author-reviewer discussion phase is nearing its conclusion (less than 24 hours remaining), we kindly ask for your feedback on our responses and the extensive revisions made to our paper ([view revised version](https://openreview.net/pdf?id=du9reSRIo1)). If our revisions adequately address your concerns, we would greatly appreciate it if you could reconsider your evaluation.
> > >
> > > Should any additional questions or concerns arise, we are more than happy to provide further clarifications. Thank you for your time and thoughtful consideration.

---

### Official Review · Reviewer_KZPo · 2024-11-03

**Soundness:** 3
**Presentation:** 3
**Contribution:** 2
**Rating:** 6
**Confidence:** 3

**Summary:**

A foundation model for vehicle routing problems was presented in the manuscript. Mixed training, global feature embedding in a transformer were proposed to increase the training advantage. The foundation model is better than MTPOMO, MVMoE and AL in uniform data, while the improvement in CVRPLIB is quite marginal.

**Strengths:**

The foundation model is better than MTPOMO, MVMoE and AL in uniform data. The normalization of rewards and layers makes a lot of sense to standardize the training of different vehicle routing problems that take on different features. Extending zero-shot and few-shot capabilities of a foundation model advances the frontier of large optimization models.

**Weaknesses:**

The foundation model is not validated comprehensively. The zero-shot validation for a mere M constraint is not convincing as it is very relevant to constraints in training batch. More constraints in zero-shot validation are supposed to make a more solid work. The zero-shot didn't account for MTPOMO, MVMoE that were only compared in Table 1. Besides uniform data, the foudanation model is validated in CVRPLIB ignoring more practical vehicle routing problems like soloman VRPTW data. The improvement is incremental and still fall behind HGS-VRP.

**Questions:**

1. Could authors explain their rationale for selecting the mixed backhaul (M) constraint for zero-shot validation, rather than other practical constraints like multiple depots? Please discuss any other choices of validation task related to real-world VRP applications.

2. What is training time of the foundation model, MTPOMO, MVMoE? MTPOMO and MVMoE are retrained by the same data and same training resource, including dataset and training duration?

3. How does the difference of AL and EAL influence outcome, as EAL keeping a predefined zero matrix seems not to influence AL that puts in the matrix in later phase? It's better to include empirical results or visualizations showing the effects of these different approaches at various stages of training.

4. Please discuss any potential trade-offs between the two methods and situations where one might be preferable over the other.

---

> ### Author Response · Authors · 2024-11-25
>
> We thank you for your effort in reviewing our paper and providing insightful feedback! We will provide detailed responses, where `W` denotes weaknesses and `Q` denotes questions.
>
> **W1/Q1. Why did you choose mixed backhauls to evaluate unseen variants? Adding new variants such as multi-depots would make for a more solid work.**
>
> The rationale behind initially choosing mixed backhauls is that mixed backhauls substantially change the solution generation process: The model needs to keep track of the available backhaul load at each step, i.e., the difference between the vehicle capacity and the used backhaul capacity and carefully plan to have enough capacity to load the backhauls.
>
> However, we do agree with you that adding new variants would make for a much more interesting experiment. Thanks to your suggestion, we have added multi-depot variants during the rebuttal, doubling the number of variants studied in our paper from 24 to 48 (see updated Table A.1, with updated environment details throughout Appendix A). To the best of our knowledge, ours is the first method whose environment and model can represent this many VRP variants.
>
> We then studied both the zero-shot generalization and finetuning performance across three different settings: 1) unseen mixed backhauls (MB), which is the setting from the main text and adds 8 new variants 2) unseen multi-depots (MD), which adds 16 new variants and 3) unseen mixed backhauls and multi-depots (MB+MD) which is the hardest setting for zero-shot generalization and finetuning, adding 8 additional variants. In total, from the original 16 variants we pretrained our models on, we generalize and finetune to 32 new variants.
>
> The newly added Appendix D.3 shows our results. RouteFinder models perform the best in zero-shot generalization across all experiments; moreover, EAL finetuning achieves the same zero-shot performance as the backbone RouteFinder model thanks to the zero-padded initialization, while AL does not due to the introduction of untrained embedding layers. Notably, experiments with multiple depots are much harder than mixed backhaul variants since they require the model to understand multiple starting (and returning) point locations and to schedule vehicle assignments to their respective depots efficiently. EAL performs the best across all variants in finetuning performance. Remarkably, EAL’s performance compared to AL and retraining a model from scratch is more prominent with the increasing difficulty of the finetuning task from MB to MB+MD, indicating it is a suitable method for efficient deployment in finetuning to new tasks.
>
> **W2. MTPOMO and MVMoE were not compared in zero-shot.**
>
> We have added the zero-shot results for MTPOMO and MVMoE on new tasks according to our above answer to W1 in Tables D1, D2 and D3. Overall, RouteFinder shows the best zero-shot generalization performance on unseen tasks.
>
> **W3. Lacking comparisons in non-uniform data.**
>
> As you mentioned, we compared RouteFinder and multi-task as well as the single-variant POMO in CVRPLIB, which demonstrates exceptional generalization ability of our model across real-world large-scale datasets. We also added new out-of-distribution comparisons for single-variant POMO, multi-task baselines, and RouteFinder models across three different scenarios: 1) unseen values of capacities 2) unseen values of time windows and 3) unseen values of distance limits. Overall, we can see that RouteFinder is robust, and its advantage compared to single-variant models is more pronounced the further we get from the training distribution.
>
> **W4. The model still falls behind HGS.**
>
> This is true, at least for the studied VRP variants. However, a motivation for our work is that not all VRP variants have strong traditional solvers available. HGS results from decades of VRP research on heuristic methods, and outstanding efforts are required to model new variants outside of the studied ones. RouteFinder is a promising direction because it is not only generally faster but can also automatically discover heuristics, unlike traditional solvers, essentially transforming a complex VRP into a data science problem. Given its strong finetuning ability, we expect practitioners to be able to model new VRPs that cannot be easily modeled by traditional solvers. Another motivation is that usually, especially on larger scales, HGS is much slower than learned VRP solvers, which motivates the creation of models such as ours. Neural combinatorial optimization is a quickly evolving field, and we believe that learning-based methods will eventually surpass traditional solvers and/or work alongside them to solve complex VRPs. In this sense, we believe RouteFinder is a notable step towards this final goal.

---

> > ### Author Response · Authors · 2024-11-25
> >
> > **Q2. What are the resources used to train RouteFinder and baselines?**
> >
> > This information is available in Section 5, which we report here: "*All training runs are conducted on NVIDIA A100 GPUs and take between 9 to 48 hours per model. Evaluation runs are conducted on an AMD Ryzen Threadripper 3960X 24-core CPU with a single RTX 3090 GPU.*". All experiments ensure fairness by employing, for example, the same number of steps, learning rate, batch size, and resources. These are available in Section 5 under the "training" paragraph, with more details available in Appendix C.1. Appendix A.1 also describes the data generation procedure, which the same across all models.
> >
> > **Q3. How do AL and EAL differ?**
> >
> > EAL differs from AL in that it keeps the first initial $k$ linear projection rows the same, hence inheriting them from the pretrained model. To ensure compatibility with a new task with $k+l$ raw features (with $l$ new features) we add $l$ new rows initialized with zeros. This has an important property. Suppose we trained the model on a multitude of variants, including CVRP. If we use AL, we lose the information of the pretrained linear projections for CVRP; hence the model will not be able to construct CVRP solutions unless finetuning is applied. On the contrary, EAL keeps this information, which can be _transferred_ for new tasks. For example, for a multi-depot VRP, we would like to keep the information we learned about depot locations since they are important for VRPs.
> >
> > This means that EAL, in practice, can keep the zero-shot performance of a pretrained model when applied to a new task. We added more evidence for this effect throughout new experiments in Tables D1 to D3, and also showed the importance of EAL in new Figure D.6 across epochs, where it can be seen how EAL yields fast finetuning to new variants.
> >
> > **Q4. What are the potential trade-offs between AL and EAL?**
> >
> > AL might be preferable to EAL only if the new tasks do not share any raw feature with the previously seen tasks. However, in the context of VRPs, this would rarely happen since features like locations and demands are required in the very definition of VRPs. Thus, in Foundation Models for VRPs, EAL finetuning would almost always be preferable to AL.

---

> > > ### Author Response · Authors · 2024-12-02
> > > **Request for Feedback**
> > >
> > > As the author-reviewer discussion phase is nearing its conclusion (less than 24 hours remaining), we kindly ask for your feedback on our responses and the extensive revisions made to our paper ([view revised version](https://openreview.net/pdf?id=du9reSRIo1)). If our revisions adequately address your concerns, we would greatly appreciate it if you could reconsider your evaluation.
> > >
> > > Should any additional questions or concerns arise, we are more than happy to provide further clarifications. Thank you for your time and thoughtful consideration.

---

### Official Review · Reviewer_X8vs · 2024-11-04

**Soundness:** 2
**Presentation:** 2
**Contribution:** 2
**Rating:** 5
**Confidence:** 4

**Summary:**

This paper proposes a unified approach called RouteFinder to model different Vehicle Routing Problem (VRP) scenarios using a transformer-based encoder and global attribute embeddings. This approach provides a scalable solution for diverse routing challenges. Mixed Batch Training (MBT) and Multi-Variant Reward Normalization are introduced to enable efficient training across multiple problem variants. Specifically, Multi-Variant Reward Normalization balances reward scales to ensure stable learning. Efficient Adapter Layers (EAL) enhance the model's adaptability, allowing it to handle new VRP variants with unseen attributes without full retraining. This paper includes extensive experiments that demonstrate the competitive performance of RouteFinder compared to state-of-the-art baselines.

**Strengths:**

1. The introduction of RouteFinder as a foundation model for various VRP variants is a significant step forward, offering flexibility and reducing the need for specialized models for each variant. This can lead to more accessible and scalable VRP solutions across diverse scenarios.

2. The model employs a transformer-based encoder with Global Attribute Embeddings, improving the understanding and differentiation of VRP tasks, which translates to better adaptability and solution quality.

3.  Mixed Batch Training (MBT) allows RouteFinder to train on multiple VRP variants simultaneously, supporting diverse distributions and ensuring model stability. The Multi-Variant Reward Normalization further strengthens robustness, addressing reward scale differences effectively across VRP types.

4.  EAL enables fine-tuning of the pre-trained RouteFinder model to handle new VRP variants, providing a lightweight and promising solution for unseen attributes. This feature is especially advantageous in real-world applications, where new constraints may arise.

5. Extensive experimentation on 24 VRP variants showcases the competitive performance of RouteFinder, indicating its effectiveness and superior performance compared to recent baselines in multi-task VRP solutions.

**Weaknesses:**

1.  Although RouteFinder demonstrates strong generalization across VRP variants, some performance trade-offs arise when compared to models designed for specific problem variants. Additional analysis could help address these trade-offs, especially in large-scale deployments.
2.  While MBT is an efficient training method, the paper could provide a more detailed examination of its limitations, particularly in cases of significantly imbalanced attribute distributions. Including more ablation studies might clarify MBT’s optimal conditions.

3.The ablation studies primarily focus on individual components. Exploring potential interactions among these components might offer insights into possible synergistic or conflicting effects, which could enhance the model's overall efficiency.

4. The use of transformers and attribute composition may make RouteFinder somewhat of a "black box." Adding interpretability methods, like attention visualization, could improve the practical usability and trustworthiness of the model by shedding light on its decision-making processes.

**Questions:**

1. It seems there is not yet a neural method that could be trained on a single VRP variant or task and outperform the strong traditional method like LKH, HGS or SISR on that task, then what is the point in training a unified model across multiple tasks with much performance drop?
2. This paper is quite similar to 'RouteFinder: Towards Foundation Models for Vehicle Routing Problems, ICML Workshop'. And what is the major difference?
3. In the main body, only O, L and M are considered as the global attribute. However, in appendix, late time window and depot location are also considered as global attribute. Why there is such an inconsistence?
4. The authors claim that RMS Norm could speed up convergence, SwiGLU could help model the data relationship. However, those claims are not fully supported in the paper.
5. In Table 5.1, the objective value of VRPBLTW is 350.808. Please elaborate what happened to the model that leads to such an inferior result.

---

> ### Author Response · Authors · 2024-11-25
>
> We sincerely thank you for your time and thoughtful feedback in reading our paper! We will provide detailed responses, where `W` denotes weaknesses and `Q` denotes questions.
>
> **W1. Performance trade-offs when compared to models designed for specific problem variants are not studied.**
>
> Indeed, training a model on a single variant may yield better results in-distribution. However, RouteFinder is much more robust than models trained on single variants when evaluating out-of-distribution cases. For example, on large-scale CVRPLIB instances (Table D.8 in Appendix), RouteFinder is much better than POMO trained specifically for CVRP. We added new experiments in Section D.4, where we compare  the single-variant POMO trained on specific variants such as CVRP, VRPL and VRPTW and compare the results with RouteFinder on out-of-distribution cases for instances with capacities, time windows, and distance limits. Interestingly, we find that while POMO trained on specific variants can perform better close to the training distribution as expected, it performs worse out-of-distribution, whereas RouteFinder demonstrates exceptional generalization.
>
> **W2. Missing analysis on MBT limitations for significantly imbalanced distributions.**
>
> To address this, we conducted new experiments to analyze MBT's performance under significantly imbalanced attribute distributions available in Section D.2 of the Appendix. Using our RouteFinder model with and without MBT, we ran training runs at different sampling probabilities of time-window attributes $p_\text{tw} = \{  0.5, 0.25, 0.10\}$ to examine different levels of imbalance.
> The results (Figure D.5) show that as $p_\text{tw}$ decreases, convergence slows for challenging variants like VRPTW, while majority-dominant variants (e.g., CVRP) converge faster. Crucially, MBT stabilizes training across all settings, particularly benefiting minority variants by mitigating instability and ensuring smoother, faster convergence. This demonstrates that MBT is a robust strategy for improving training stability and convergence in imbalanced scenarios.
>
>
> **W3. The paper does not show the interaction among components.**
>
> Figure 5.1 shows the ablation studies of several RouteFinder's components alongside a sensitivity analysis on Reward Normalization. All the proposed components significantly improve the performance of our model. We also added additional ablation studies, especially regarding the Transformer Encoder components in Appendix D.1. Here, we compare RouteFinder with no SwiGLU and no RMSNorm, with SwiGLU only, with RMSNorm only, and with both (full Transformer Encoder). The results demonstrate that the Tranformer Encoder performs best (more in the answer to Q4).
>
>
>
> **W4. Issues in interpretability due to transformers.**
>
> We agree that understanding a model’s decision-making process is critical for practical usability and trustworthiness. While we currently focus on addressing the core research challenges of learning and optimization in complex combinatorial tasks, integrating interpretability methods like attention visualization is indeed a promising direction. To shed some light on this, we added new experiments visualizing the hidden space via t-SNE between transformer layers across all models, to understand how RouteFinder and previous models understand different attributes (Figures D.7 and D.8 in Appendix). We note that RouteFinder exhibits more complex patterns the deeper the layers. Interestingly, when analyzing the hidden space on a _per attribute_ basis, our model yields well-defined, tightly grouped, and distinct clusters on all variants, which is a strong
> indicator of its capability to generalize and specialize effectively in solving diverse VRP variants. For example, in contrast to the baselines, RouteFinder effectively separates time window variants into two distinct clusters — those with open routes and those without — enabled by the Global Attribute Embeddings.

---

> > ### Author Response · Authors · 2024-11-25
> >
> > **Q1. There is still no neural method that can outperform SOTA heuristics on single variants. Why train a foundation model on multiple variants with some performance drop?**
> >
> > Thank you for your question — this is indeed fundamental to our motivation. There are several reasons behind our research.
> >
> > For one, foundation models generally demonstrate better generalization performance. We studied this insight across sizes (see Appendix D.5) and distributions (see new experiments we added in Appendix D.4), where RouteFinder can consistently outclass problem-specific methods, even when it has seen just a fraction of the instances for that task.
> >
> > Another motivation for our work is that not all VRP variants have strong traditional solvers available. HGS results from decades of VRP research on heuristic methods, and outstanding efforts are required to model new variants outside of the studied ones. RouteFinder is a promising direction because it is not only generally faster but can also automatically discover heuristics unlike traditional solvers, transforming a complex VRP in essentially a data science problem.
> >
> > Moreover, foundation models as RouteFinder can be easily finetuned — including with our proposed EAL — as demonstrated in our experiments, including the new ones we added on multi-depot variants (Appendix D.3) which bring up the count of studied variants in this work to 48. In real-world scenarios, there may be hundreds of combinations of different variants. Training a model once and finetuning to new variants can be much more efficient than the daunting process of training models for several variants from scratch. For example, to obtain models for just our 16 initial pre-training variants, would require around $16\times$ our compute budget.
> >
> >
> > Finally, just as foundation models in areas like LLMs are transforming industries by scaling and outperforming task-specific models in downstream tasks, we anticipate that future work, including upcoming versions of RouteFinder, will eventually surpass traditional methods.
> >
> >
> >
> >
> > **Q2. What are the differences between your paper and another similar workshop paper?**
> >
> > Without violating anonymity, we kindly note that workshop papers are preliminary unpublished work, so they should not be relevant during the reviewing process.
> >
> >
> > **Q3. Why do you consider late time window and depot location as global attributes in the Appendix?**
> >
> > In practice, we model global attributes by projecting them on the depot node embeddings. Since global attributes contain a single value for all problem nodes, we embed them in depot nodes, in a similar fashion to how traditional solvers as PyVRP encode information about the global problem structure on depot nodes. We clarify this in Appendix B.2.
> >
> > **Q4. Can you provide evidence on the effect of the Tranformer Encoder components?**
> >
> > Thank you for pointing out the previous lack of sufficient evidence on this. We added new experiments in Appendix D.1 studying Transformer Encoder components for RouteFinder with no SwiGLU and no RMSNorm, with SwiGLU only, with RMSNorm only, and with both (full Transformer Encoder), with results demonstrating that the Tranformer Encoder peforms best in Figure D.1. Moreover, we have added a plot showing the validation gaps over epochs that provide insights on the importance of interaction between the proposed components in Figure D.2. While the final performance for the variant with only SwiGLU outperforms the vanilla baseline due to its enhanced capability in representation learning, its convergence is slower in the beginning. However, the full Transformer Encoder containing both RMSNorm and SwiGLU not only performs the best, but also converges the fastest, indicating the importance of interactions of each single component.
> >
> > **Q5. An objective value of VRPBLTW is 350.808 in Table 1; what happened to the model?**
> >
> > We mistyped an extra "5"; the objective value is 30.808, while the gap was already correct. We have now corrected this value in Table 1. Thank you for spotting this typo.

---

> > > ### Author Response · Authors · 2024-12-02
> > > **Request for Feedback**
> > >
> > > As the author-reviewer discussion phase is nearing its conclusion (less than 24 hours remaining), we kindly ask for your feedback on our responses and the extensive revisions made to our paper ([view revised version](https://openreview.net/pdf?id=du9reSRIo1)). If our revisions adequately address your concerns, we would greatly appreciate it if you could reconsider your evaluation.
> > >
> > > Should any additional questions or concerns arise, we are more than happy to provide further clarifications. Thank you for your time and thoughtful consideration.

---

> > > ### Comment · Reviewer_X8vs · 2024-12-03
> > > **The proposed learning-based VRP solver shows improvements over existing neural solvers but lacks the performance and impact to surpass HGS or HGS-PyVRP, limiting its significance.**
> > >
> > > I appreciate the authors' efforts in the rebuttal. However, in my view, if the learning-based VRP solver cannot outperform HGS in a single-task setting, it is unlikely to achieve superior performance to HGS-PyVRP in a multi-task setting. While the authors have made progress in improving upon existing neural solvers like MTPOMO and MVMOE (I do not see much impact in doing this), their approach, seems without any chance, to surpass HGS, which limits its impact. At the same time, although the authors highlighted the solver's adaptability to various VRP variants as a strength, it is worth noting that HGS, which initially focused on CVRP and CVRPTW, has since evolved to address dozens of VRP variants with much better solution quality compared to RF, MVMOE, and MTPOMO. Meanwhile, considering comments from other reviewers, I would like to maintain my initial score.

---

> > > > ### Author Response · Authors · 2024-12-03
> > > > **Do neural solvers need to beat HGS to be valuable?**
> > > >
> > > > Thanks for your answer.
> > > >
> > > > We would like to clarify our perspective further.
> > > >
> > > > > The authors have made progress in improving upon existing neural solvers like MTPOMO and MVMOE [...]  It is worth noting that HGS, which initially focused on CVRP and CVRPTW, has since evolved to address dozens of VRP variants with much better solution quality compared to RF, MVMOE, and MTPOMO.
> > > >
> > > > Some comments:
> > > > 1. MTPOMO and MVMoE, as RouteFinder, can automatically discover heuristics; HGS requires much more effort in manual tuning and implementation, given new requirements.
> > > > 2. The goal of much constructive Neural Combinatorial Optimization (NCO) is not necessarily to find better final solutions or provably optimal but to obtain fast heuristics at a fraction of the cost. In this sense, we believe RouteFinder can be a valuable tool.
> > > > 3. Other works in foundation models for combinatorial problems (such as [this concurrent work](https://openreview.net/forum?id=z2z9suDRjw)) also do not beat the final HGS performance, as motivations include fast adaptability to new problems and self-discovery of heuristics, much like our work.
> > > >
> > > > > I do not see much impact in [...] improving upon existing neural solvers like MTPOMO and MVMOE.
> > > >
> > > > While we respect this opinion, we believe this assessment is rather subjective. We want to note that since RouteFinder is a research work in RL for VRPs, the idea is to improve upon state-of-the-art results for neural methods. We believe we achieved that.
> > > >
> > > > We would also like to note that **neither MTPOMO nor MVMoE beat HGS and were published in KDD 2024 and ICML 2024**, respectively, and have already gathered a few citations, speaking to the value of this research area.
> > > >
> > > > ---
> > > >
> > > > To answer our initial question: "**Do neural solvers need to beat HGS to be valuable?**", we will report the reviewer guidelines available [here](https://iclr.cc/Conferences/2025/ReviewerGuide#:~:text=Q%3A%20If%20a%20submission%20does%20not%20achieve%20state%2Dof%2Dthe%2Dart%20results%20%2C%20is%20that%20grounds%20for%20rejection%3F):
> > > >
> > > > > Q: If a submission does not achieve state-of-the-art results , is that grounds for rejection?
> > > >
> > > > > A: **No, a lack of state-of-the-art results does not by itself constitute grounds for rejection**. Submissions bring value to the ICLR community when they convincingly demonstrate new, relevant, impactful knowledge. Submissions can achieve this without achieving state-of-the-art results.
> > > >
> > > >
> > > > Finally, we note that **RouteFinder achieves state-of-the-art results in multi-task learning for VRPs**, and given our contributions, we believe our work will enable several future works.
> > > >
> > > > We sincerely hope you can understand our position — we might add, in the name of the NCO community — and we hope you may reconsider your opinion in light of the above discussions.
> > > >
> > > > Best,
> > > >
> > > > RouteFinder Authors

---

### Author Response · Authors · 2024-11-25
**General Response**

We would like to thank all reviewers and the Area Chair for their time and effort in evaluating our work!

We are pleased to see that the reviewers recognized RouteFinder as a meaningful step forward in foundation models for VRPs. Specifically, the reviewers highlighted that RouteFinder is a **significant advancement** in modeling VRP variants (`X8vs`, `KZPo`) representing a **strong attempt at generalization** (`kTqa`), introduces **novel methodologies** such as Mixed Batch Training (MBT) and Efficient Adapter Layers (EAL) (`X8vs`, `KZPo`, `p8aH`), and demonstrates **strong empirical results** across a wide range of VRP variants (`X8vs`, `kTqa`, `p8aH`).

During the rebuttal period, we carefully considered all reviewers' feedback and conducted **extensive new experiments**:

1. The unified VRP environment can now also model multi-depot VRP, **increasing the number of variants from 24 to 48**, making RouteFinder the first neural approach in the literature to model this many variants
2. **New EAL experiments** evaluating zero-shot and fine-tuning performance, including both previous and new multi-depot variants, with a detailed analysis of how the model's performance evolves over fine-tuning epochs
3. **Ablation study on Transformer Encoder** demonstrating the importance of each component
4. **MBT experiments on imbalanced attribute distributions** demonstrating how MBT is important especially for minority samples
5. **New experiments on out-of-distribution attribute values** that show how RouteFinder can outperform  models trained specifically for a single-variant
6. Experiments on **EAL finetuning on single-variant models** that show RouteFinder achieves better performance in finetuning to unseen variants compared to single-variant models
7. New **t-SNE latent space experiments** demonstating the importance of the Global Attribute Embeddings
8. More **MBT experiments and plots** that showcase increased training stability and better performance

We have revised multiple sections of the manuscript in response to the reviewers' constructive feedback. For convenience, the new and updated content has been highlighted in blue in the revised PDF.

---

We hope our responses can address all concerns, and we welcome further discussions if any point is unclear.

RouteFinder Authors

---

### Author Response · Authors · 2024-12-02
**Request for Feedback**

Dear Reviewers,

Thank you all for your thoughtful and constructive reviews of our submission. We deeply appreciate the time and effort you have dedicated to providing valuable feedback.

In response to your comments, we have worked hard and made substantial updates to the paper, including conducting extensive additional experiments, such as expanding the paper to 48 VRP variants and providing further clarifications that directly address the concerns raised. We believe these revisions significantly strengthen the work, and we hope that the changes now address the issues raised in your reviews.

Given the extent of the revisions, we kindly ask that you take a moment to re-evaluate the revised version of the paper. If the updates have addressed your concerns, we would greatly appreciate it if you could reconsider your evaluation. Your feedback is crucial in ensuring a fair and thorough review process, and we want to ensure the paper is assessed based on its updated and improved version.

If there are any remaining concerns or if additional clarification is needed, we are more than happy to provide further details or make any necessary adjustments.

Thank you once again for your time, effort, and continued support in this process.

Best regards,

RouteFinder Authors

---

### Author Response · Authors · 2024-12-04
**Final Message**

We thank the reviewers once more for their initial feedback. We will provide a summary of our paper and the rebuttal.


## RouteFinder's Core Contributions

RouteFinder is a foundation model for vehicle routing problems (VRP). Our contributions include:

1. **Unified Framework for Multiple VRP Variants:** We introduce a general framework that can solve various VRP variants by handling any number of attributes, making it applicable to a wide range of problem settings.

2. **Transformer-Based Architecture with Global Attribute Embeddings:** Our approach uses a modern Transformer architecture and Global Attribute Embeddings, allowing the model to understand and differentiate between VRP variants more effectively.

3. **Novel RL Techniques:** We propose two novel techniques—**Mixed Batch Training (MBT)** and **Multi-Variant Reward Normalization**—to ensure stable and effective training across multiple VRP variants.

4. **Efficient Adapter Layers (EAL):** We introduce a lightweight yet powerful mechanism for fine-tuning pre-trained models to tackle new variants with unseen attributes.

We evaluated RouteFinder across 48 VRP variants -- the first neural model to do so -- demonstrating that it significantly outperforms recent multi-task learning models.


## Rebuttal Remarks

During the rebuttal period, we carefully considered all comments and conducted extensive additional experiments to address the concerns and provided further evidence of our RouteFinder's effectiveness.

While we believe we have fully addressed the reviewers' concerns, with extensive experiments and studies that added 10 additional pages to our manuscript, doubled the number of VRP variants, and took almost 2 weeks of full-time work, we regret that 3 out of 4 reviewers did not engage in discussions. While we thank the only reviewer who replied, X8vs, for his/her reply, we respectfully disagree with the assessment that beating the HGS performance for a neural network is the criterion of acceptance. We hope you understand our position.

---

We respectfully request that the reviewers reconsider their decision in light of the clarifications and additional evidence provided. We hope for a fair reevaluation of our work.


Best Regards,

RouteFinder Authors

---

### Meta-Review · Area_Chair_1rN9 · 2024-12-23

**Metareview:**

The submission proposes RouteFinder, a framework for training netural networks for Vehical Routing Problems (VRPs). Key claimed contributions include - 1) Transformer encoder and global attribute embeddings, 2) Mixed batch training, 3) Multi-variant reward normalization, and 4) efficient adapter layers for finetuning models for new problems.

The submission initially received ratings of 5, 5, 5, and one reviewer increased their rating to 6 after the rebuttal. The ACs have taken into account the concerns of the authors regarding certain reviews.

Based on the reviews, discussions, and readings of all submitted material by the ACs, the claimed contributions fall short, as they heavily borrow from prior work:
- The transformer architecture is borrowed from prior work including the Llama LLM, and is not novel in its design. All the described components are commonly used (MHA, RMSNorm, SwiGLU, Flash attention, etc.) and there are no novel findings here.
- Mixed Batch training: Authors claim that prior work samples a particular instance of the VRP problem in each training batch, potentially hindering training. A natural baseline would be gradient accumulation across multiple batches, where each batch contains a single variant. Note that transformer architectures do not have batchnorms and different samples never interact with each other. By accumulating gradients across multiple varied batches, you can simulate a mixed batch. This was however not compared against.
- The efficient adapter layers (EAL) in this work expand linear projection layers by appending the matrix with 0s corresponding to the new dimensions. This is however, not novel. Zero initialization of new weights is a well known technique in neural network surgery and expansion/inflation.

The above reasons greatly reduce the novelty of this work, bringing it below the bar for acceptance. The ACs do not find sufficient cause to overturn the negative consensus of the reviewers.

**Additional Comments On Reviewer Discussion:**

Multiple reviewers raised questions regarding the value of training neural networks for the VRP tasks where traditional solvers still dominate. The ACs note that this is not a disqualifying factor, as long as the submission provides novel insights or contributions.

---

### Decision · Program_Chairs · 2025-01-22

Reject